



# Silicification in the Ocean: from molecular pathways to silicifiers' ecology and biogeochemical cycles

Ivia Closset[1,2], J. Jotautas Baronas[3], Fiorenza Torricella[4,5], Félix de Tombeur[6,7], Bianca T.P. Liguori[8], Alessandra Petrucciani[9,10], Natasha Bryan[11], María López-Acosta[12,*], Yelena Churakova[13], Antonia U. Thielecke[11,*], Zhouling Zhang[8], Natalia Llopis Monferrer[14,15], Rebecca A. Pickering[16], Mathis Guyomard[17], Dongdong Zhu[18]

[1]Finnish Meteorological Institute, Dynamicum Erik Palménin aukio 1, 00560 Helsinki, Finland
[2]Marine Science Institute, University of California Santa Barbara, Santa Barbara, CA, USA
[3]Department of Earth Sciences, Durham University, UK
[4]Department of Mathematics, Informatics and Geosciences, MIGe, University of Trieste
[5]Instituto of Polar Sciences, ISP-CNR, Bologna
[6]CEFE, Univ Montpellier, CNRS, EPHE, IRD, Montpellier, France
[7]School of Biological Sciences and Institute of Agriculture, The University of Western Australia, Perth, WA, Australia
[8]GEOMAR, Helmholtz Centre for Ocean Research, Kiel, Germany
[9]Dipartimento di scienze della vita e dell'ambiente, Università Politecnica delle Marche, Ancona, Italy
[10]CIRCC, Consorzio Interuniversitario Reattività Chimica e Catalisi, Italy
[11]Alfred Wegener Institute Helmholtz Centre for polar and marine research, Germany
[12]Instituto de Investigaciones Marinas (IIM), CSIC, C/Eduardo Cabello 6, 36208 Vigo, Spain
[13]Centre for Ecology and Evolution in Microbial Model Systems (EEMiS), Linnaeus University, Kalmar, Sweden
[14]Sorbonne University, CNRS, UMR7144 Adaptation and Diversity in Marine Environment (AD2M) Laboratory, Ecology of Marine Plankton team, Station Biologique de Roscoff, Roscoff, France
[15]Monterey Bay Aquarium Research Institute, Moss Landing, CA, United States
[16]Department of Geological Sciences, Stockholm University, Stockholm, Sweden
[17]LOCEAN-IPSL, Sorbonne Université (SU, CNRS, IRD, MNHN), Paris 75005, France
[18]Key Laboratory of Marine Chemistry Theory and Technology, Ministry of Education, Ocean University of China, Qingdao 266100, China

*Correspondence to*: María López-Acosta (lopezacosta@iim.csic.es) and Antonia U. Thielecke (antonia.thielecke@awi.de)





**Abstract.** The oceanic silicon cycle has undergone a profound transformation from an abiotic system in the Precambrian to a
biologically regulated cycle driven by siliceous organisms such as diatoms, Rhizaria, and sponges. These organisms actively
uptake silicon using specialized proteins to transport and polymerize it into amorphous silica through the process of
biosilification. This biological control varies depending on environmental conditions, influencing both the rate of silicification
and its ecological function, including structural support, defence, and stress mitigation. Evidence suggests that silicification
has evolved multiple times independently across different taxa, each developing distinct molecular mechanisms for silicon
handling. This review identifies major gaps in our understanding of biosilicification, particularly among lesser-known
silicifiers beyond traditional model organisms like diatoms. It emphasizes the ecological significance of these underexplored
taxa and synthesizes current knowledge of molecular pathways involved in silicon uptake and polymerization. By comparing
biosilicification strategies across taxa, this review calls for expanding the repertoire of model organisms and leveraging new
advanced tools to uncover silicon transport mechanisms, efflux regulation, and environmental responses. It also emphasizes
the need to integrate biological and geological perspectives, both to refine palaeoceanographic proxies and to improve the
interpretation of microfossil records and present-day biogeochemical models. On a global scale, silicon enters the ocean
primarily via terrestrial weathering and is removed through burial in sediments and/or authigenic clay formation. While open-
ocean processes are relatively well studied, dynamic boundary zones – where land, sediments, and ice interact with seawater
– are nowadays recognized as key regulators of silicon fluxes, though they remain poorly understood. Therefore, special
attention is given to the role of dynamic boundary zones such as the interfaces between land and ocean, the benthic zone, and
the cryosphere, which are often overlooked yet play critical roles in controlling silicon cycling. By bringing together cross-
discipline insights, this review proposes a new integrated framework for understanding the complex biological and
biogeochemical dimensions of the oceanic silicon cycle. This integrated perspective is essential for improving global silicon
budget estimates, predicting climate-driven changes in marine productivity, and assessing the role of silicon in modulating
Earth's long-term carbon balance.

# 1 Introduction

Silicon (Si) is the second most abundant element in the Earth's crust after oxygen and plays a crucial role in a variety of
biological and biogeochemical processes (Struyf et al., 2009; Tréguer et al., 2021). It is an essential nutrient used by various
terrestrial organisms, including plants and mammals, as well as a diverse range of marine life, such as unicellular diatoms and
multicellular sponges (DeMaster, 2003; Farooq and Dietz, 2015). These organisms, collectively referred to as silicifiers,
produce siliceous structures (also known as biogenic silica, bSi, or opal) through a biologically controlled process known as
biosilicification. These silicifiers modulate Si cycling in the ocean through Si uptake and as vessels of Si sedimentation and
burial. While diatoms have received most attention in biosilicification research due to their ecological importance, abundance
and relative ease to maintain in culture, they are not the only marine silicifiers. Recent studies have highlighted the importance
of other key groups such as Rhizaria and sponges (Llopis Monferrer et al., 2020; López-Acosta et al., 2018; Maldonado et al.,



2020). Silicon also contributes to other forms of biomineralization beyond siliceous structures made by silicifiers. It plays a role in the calcification of some coccolithophores (Durak et al., 2016; Ratcliffe et al., 2023a) and in the formation of silicified mouthparts in copepods (Naumova et al., 2015). In plants, Si is found at a wide range of concentrations and silicification contributes to the regulation of numerous biotic and abiotic stresses (Currie and Perry, 2007). Furthermore, the use of Si extends beyond eukaryotes, and it has been demonstrated that marine picocyanobacteria are able to accumulate important quantities of bSi (Baines et al., 2012). This growing body of work reveals that biosilicification is a widespread and evolutionary diverse phenomenon. Broadening our focus beyond diatoms is therefore essential to fully capture the complexity and global significance of biosilicification and its influence on the Si cycle in the ocean.

Particularly among non-model taxa such as protists, sponges, and picocyanobacteria, the physiological mechanisms underlying Si uptake, transport, and polymerization are still unknown, limiting our ability to generalize across taxa or predict functional responses to environmental change. Moreover, the role of silicifiers in shaping Si dynamics in the modern ocean and over geological timescales remains difficult to quantify. This is due to limited in situ measurements, unknown isotopic fractionation pathways, and the insufficient integration of biological diversity into global biogeochemical models. This includes a limited representation of how silicifiers influence other biogeochemical cycles, particularly the carbon cycle, despite their major role in trapping carbon in surface waters and contributing to its long-term sequestration into the deep ocean via the biological pump (Laget et al., 2024; Ragueneau et al., 2006; Tréguer et al., 2018). Together, these knowledge gaps hinder our ability to assess how the Si cycle may respond to ongoing environmental changes, including global warming, ocean acidification, and shifts in nutrient availability.

In celebration of the Ocean Science Jubilee, we — a group of early-career researchers specializing in biosilicification and Si cycling — have collaborated to produce this review, which synthesizes current understanding of biosilicification across diverse organisms and its influence on the global marine Si cycle. We summarize existing knowledge on Si transport and polymerization at the cellular and molecular levels, the biological functions of bSi, and the environmental and ecological factors that regulate biosilicification. We also explore the taxonomic diversity of marine silicifiers and assess their roles in Si cycling across geological timescales and the modern ocean. Where relevant, we include examples from terrestrial ecosystems to highlight similarities and differences with marine silicifiers and identify opportunities for cross-system knowledge exchange. In addition, we examine key marine Si cycling processes in the different Earth interface zones (i.e., land-ocean, sediment-ocean, and ice-ocean), which are often underrepresented in global models but play a pivotal role in Si transformation and fluxes. In doing so, we highlight how stable ($^{29}$Si and $^{30}$Si) and radioactive ($^{32}$Si) isotopic tools have been instrumental in uncovering the contributions of these interfaces, as well as the broader biogeochemical impacts of silicifiers across time and space. Finally, we identify major knowledge gaps and unknowns across taxonomic, physiological, ecological, and biogeochemical domains, and propose a framework for future interdisciplinary research to address these challenges and advance our understanding of biosilicification and the global marine Si cycle.





## 2. Silicification: a widespread yet enigmatic process

Silicification involves the incorporation of inorganic Si, primarily as orthosilicates, into living organisms to form silica

structures. This form of biomineralization represents a remarkable feat of biological engineering, characterized by the precise precipitation of amorphous silica within specialized cellular compartments. In the ocean, Si is found in dissolved form as silicic acid, which is readily available for biological uptake by organisms.

The silicification process in marine environments occurs at relatively low temperatures, typically ranging from a few degrees Celsius in deep waters or high latitudes to around 30°C in tropical surface waters. In contrast, synthetic silica formation

in industrial or laboratory settings generally requires significantly higher temperatures and controlled conditions to facilitate precipitation and structural formation (De Tommasi et al., 2017). This stark difference highlights the efficiency of biological silicification, which occurs under ambient environmental conditions without the need for high temperatures or external catalysts.

Understanding this process reveals the mechanisms by which organisms control silica formation at the nanoscale, leading

to the production of highly intricate and functionally optimized structures (see **Box 1** for tools used to study biosilicification and Si uptake). Studying biological silicification not only deepens our knowledge of biomineralization and evolutionary adaptations but also has potential applications in biomimetic materials, nanotechnology, and sustainable engineering. Despite its broad distribution among marine lineages and the well-recognized benefits, such as protection from predation, light regulation, and enhanced structural support (Ghobara et al., 2019; Pančić et al., 2019; Petrucciani et al., 2022a), the specific

requirements for Si and the mechanisms underlying its metabolism remains poorly defined in most living groups.

In marine environments, most work on silicification processes has focused on diatoms, admittedly due to their abundance and the relative ease of culturing them. In these organisms, silicification begins with the intracellular accumulation of silicate, which must reach a sufficiently high concentration (19-340 mM) for deposition (Kumar et al., 2020a). Dissolved silica (dSi) from the environment is transported across the plasma membrane into the cell and concentrated in specialized intracellular

compartments called silica deposition vesicles (**Fig. 1**; Martin-Jézéquel et al., 2000), where controlled amorphous silica deposition occurs. Given the limited availability of dSi in modern oceans (ranging from under 2 µM in the surface water of central gyres to around 100 µM in polar regions; Tréguer et al., 1995), its uptake is considered the key step in silicification. While most studies have focused on dSi influx, an often-overlooked but crucial process is Si efflux from the cell, which helps prevent excessive intracellular Si accumulation that could lead to auto-polymerization and disrupt cellular function (Petrucciani

et al., 2022b). Efflux has been proposed to operate through a mechanism similar to that of the influx but in reverse, although the role of sodium in this process remains unclear (Thamatrakoln et al., 2006).





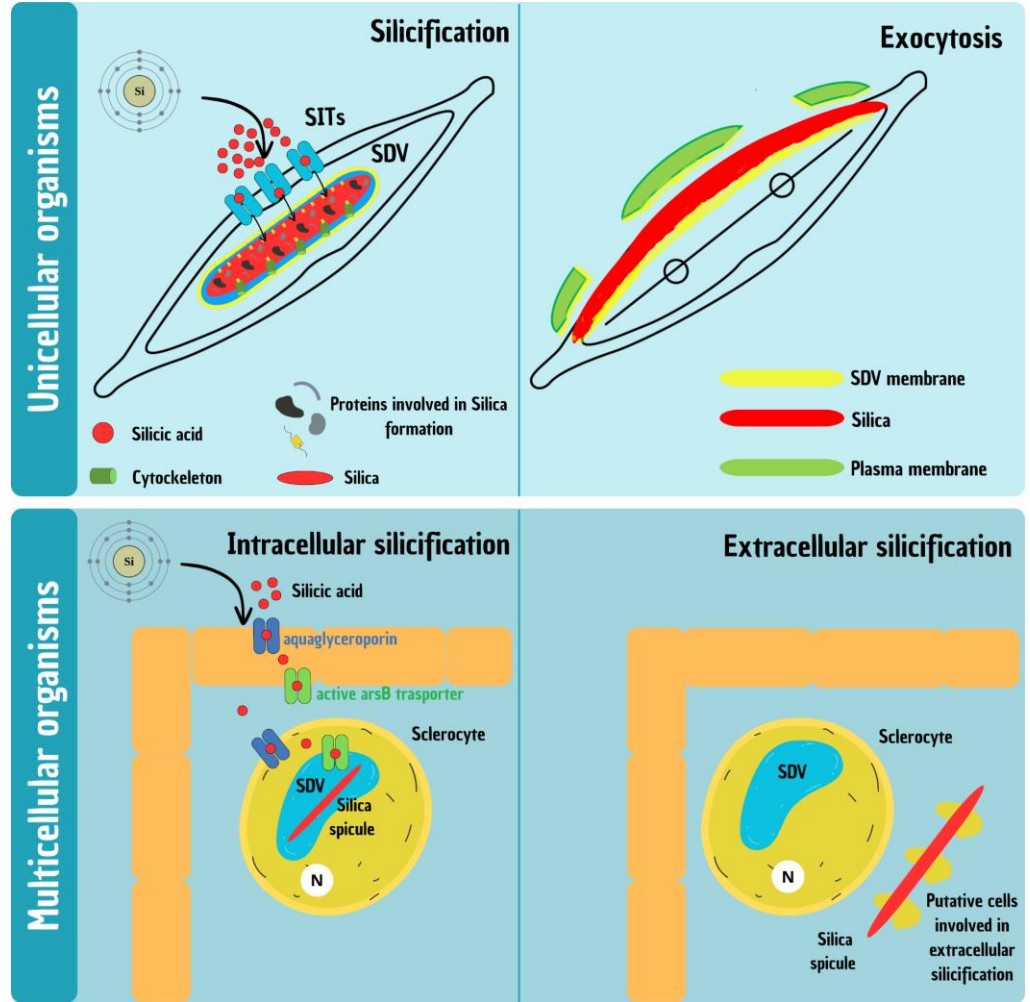

**Figure 1. Schematic representation of the silicification process in unicellular and multicellular organisms. (Top) Silicification in**
**unicellular organisms, illustrated with a model diatom. Two main phases are shown: intracellular formation (left) and exocytosis**
**(right) of the silica-based cell wall. In the first phase, dSi is transported into the membrane-bound silica deposition vesicle (SDV),**
**where polymerization forms the intricate valve structure. In the second phase, the mature silica element is exocytosed: the distal**
**SDV membrane and plasma membrane gradually detach from the newly formed silica and disintegrate in the extracellular space,**
**while the proximal SDV membrane remains and functions as the new plasma membrane, forming the interface between the cell and**
**its environment. (Bottom) Silicification in multicellular organisms, illustrated using Hexactinellid sponges. Two phases are also**
**depicted: intracellular formation (left) and extracellular elongation (right). During the first phase, dSi is transported to the silica**
**deposition vesicle (SDV) inside specialized cells called sclerocytes, where proteins and enzymes regulate silica polymerization to form**
**spicules. Once spicules outgrow the cell, formation continues extracellularly, though the molecular and cytological mechanisms**
**governing this phase remain largely unresolved.**


While silicification in unicellular organisms like diatoms is relatively well understood, the process becomes significantly more

complex in multicellular organisms. Unlike single-celled organisms, where dSi is directly taken up from the environment, in

multicellular systems, it must traverse multiple cellular barriers before reaching the site for silica deposition (**Fig. 1**). For



example, in sponges, Si uptake and deposition involve several sequential steps. Sponges take up dSi from seawater through
their outer epithelial cells and transport it internally across multiple cell layers to specialized silica-secreting cells called
sclerocytes. Within sclerocytes, dSi is concentrated in silica deposition vesicles (SDV), where polymerization occurs to form
silica spicules, the structural elements of the sponge skeleton (Maldonado et al., 2020). The formation of long spicules, known
as megascleres, is completed outside the cell, where silica is deposited externally (Schröder et al., 2007).

## 2.1 Silicon transport

Silicic acid, a small and uncharged molecule, can diffuse freely across membranes at environmentally relevant concentrations,
making diffusion the main mode of uptake in diatoms (Thamatrakoln and Hildebrand, 2008). However, the average surface
seawater silicic acid concentration in marine environments is 10 µmol $L^{-1}$ (Conley et al., 2017; Frings et al., 2016), which is
relatively low compared to the amounts required for biomineralization. To overcome this limitation, diatoms rely on
specialized protein-mediated active transport systems to efficiently take up and concentrate Si within the diatom cell, ensuring
the formation of silica-based structures. This process is tightly regulated, with specific proteins preventing premature silica
polymerization during the initial stages of silicification.

The first Si transporters (SITs) were identified in the pennate diatom *Cylindrotheca fusiformis* (Hildebrand et al., 1997).
These sodium-coupled active transporters facilitate silicic acid uptake from the environment and exhibit variations in cellular
localization, binding affinity and transport rates. Diatoms possess multiple SITs (Durkin et al., 2016) with distinct gene
expression patterns, potentially corresponding to different silicification roles (Thamatrakoln et al., 2006). Structurally, SITs
typically contain 10 transmembrane domain segments (TMDs), arranged as two symmetrical 5 TMDs, and operate through a
proposed transport model based on well-conserved EGXQ and GRQ motif pairs (Thamatrakoln et al., 2006), which are short
amino acid sequences believed to play key roles in substrate recognition and translocation across the membrane. The hydroxyl
group of silicic acid binds to the carbonyl group of the glutamine belonging to a conserved GXQ motif in the SIT proteins,
triggering a conformational change that allows silicic acid to pass through the membrane (Knight et al., 2016).

SITs have been identified in many eukaryotic supergroups, such as chrysophytes (Likhoshway et al., 2006) and
choanoflagellates (Marron et al., 2013). These SITs, along with those from diatoms, are proposed to have evolved from SIT-
like (SIT-L) proteins through duplication and fusion events (Knight et al., 2023; Marron et al., 2016). SIT-L proteins resemble
half-completed variations of SITs, containing only 5 TMDs and a single EGXQ-GRQ motif pair, and likely originated in
bacteria (Marron et al., 2016). Although SIT-Ls share structural similarities with SITs, their role in silicic acid uptake remains
uncertain. This uncertainty is reinforced by the observation that eukaryotic lineages containing SIT-Ls tend to be less silicified
than those with SITs (Hendry et al., 2018; Marron et al., 2016; Ratcliffe et al., 2023b). Recently, a SIT-L protein from the
coccolithophore *Coccolithus braarudii* was confirmed to actively drawdown silicic acid in a heterologous system (Ratcliffe et
al., 2023b). However, this study also suggests that SIT-Ls are less efficient transporters than conventional SITs, as their silicic
acid uptake remained undetectable even when using highly sensitive radioisotopic methods ($^{32}$Si) typically used to measure
uptake of silicic acid in diatoms and Rhizaria (see **Box 1**).



While SITs and SIT-Ls play a crucial role in Si transport in unicellular silicifiers, the mechanisms governing this process in multicellular organisms appear to be fundamentally different, suggesting independent evolutionary pathways for silicification. In sponges, Si transport involves both passive and active transport mechanisms. Passive transport occurs via

aquaglyceroporins, facilitating Si diffusion along its concentration gradient. Active transport is mediated by $Na^+$-coupled co-transporters, enabling Si accumulation against its gradient. This dual transport system allows sponges to acquire dSi from seawater and transport it through the various cell barriers to the sites of spicule formation, where dSi is concentrated to initiate silicification (Maldonado et al., 2020).

Research on Si transport in marine macrophytes is rather limited, despite recent evidence that they may play a key role in

marine Si biogeochemistry (e.g. macroalgae; Yacano et al., 2021). In seagrasses, a protein called Siliplant1 has been associated with the vesicular transport of silica into the apoplast, the network of cell walls and intracellular spaces outside the plasma membrane, facilitating its incorporation into the extracellular matrix (Kumar et al., 2020b; Nawaz et al., 2020). For example, in *Zostera marina*, this mechanism likely facilitates the deposition of amorphous silica within cell walls and intercellular spaces that can persist even after harsh chemical treatments like alkaline digestion (Roth et al., 2025). Despite such evidence of "active

Si transport", more research is needed to better understand and characterize Si accumulation in marine macrophytes. In this respect, research on terrestrial plant species may provide valuable information. In particular, the tremendous variation in plant Si accumulation between terrestrial plant species (e.g. in leaves, Si concentration ranges from negligible amounts to over 10% of dry weight; de Tombeur et al., 2023b) is classically attributed to the relative importance of passive and active transport (Liang et al., 2006). In short, the presence of specific influx and efflux channels encoded by specific genes would allow silicic

acid uptake from the soil solution. Notably, a gene coding for cell-specific silica deposition has been recently identified in rice (Mitani-Ueno et al., 2023), shedding light on why certain taxa accumulate significantly more Si than others (Deshmukh et al., 2020; Hodson et al., 2005). However, such gene-based mechanisms remain poorly explored in marine plants. For instance, only a few proteins, such as Siliplant1, have been tentatively linked to Si transport in seagrasses (Kumar et al., 2020b; Nawaz et al., 2020), with homologues of other known Si transporter genes in plants yet to be found.

**2.2 Silica polymerization**

Extensive research has sought to understand how silicic acid is internalized by silicifying organisms. However, the core of the silicification process lies in the polymerization of soluble silicic acid into amorphous silica, a transformation tightly regulated by biological mechanisms. This process is still being actively studied to better understand its details, even though a lot of research has already been done in model diatoms (Hildebrand et al., 2018; Mayzel et al., 2021; McCutchin et al., 2025), as

well as more recently in sponges (Shimizu et al., 2015, 2024; Wang et al., 2012), and plants (Kumar et al., 2017a, b, 2021; Zexer et al., 2023). Silicification can occur intracellularly or extracellularly, but in all cases, specialized proteins guide the formation of silica into intricate nanoscale shapes.

In diatoms, silica polymerization and cell wall synthesis occur within specialized intracellular compartments known as silica deposition vesicles (SDVs), which are enclosed by a membrane called silicalemma. These vesicles have been visualized



using transmission electron microscopy but have yet to be biochemically isolated (Hildebrand et al., 2018; Hildebrand and Lerch, 2015). The internal environment of the SDVs is slightly acidic, facilitating the polymerization of silicic acid into a gel-like silica network (Iler, 1979). Once inside the cells, yet unidentified binding components –potentially ligands or proteins– are thought to maintain silicic acid in solution and prevent premature autopolymerisation, which occurs at concentrations exceeding 2 mM (Martin-Jézéquel et al., 2000). Although polymerization was long assumed to be restricted to SDVs, recent

discoveries in certain diatom species with elaborate silica appendages suggest that silicification may also occur extracellularly (Mayzel et al., 2021).

   Silicon polymerization in diatoms is finely controlled by specialized proteins, with silaffins being the most extensively studied. These small, lysine- and serine-rich peptides catalyse silica polymerization, and the morphology of the resulting silica structure depends on their concentration and specific combination (Kröger et al., 1999, 2002). Other key proteins include long-

chain polyamines (LCPAs), which catalyse silica polymerization in the presence of polyanions, and silacidins, proteins rich in serine and acidic amino acids that catalyse silica formation in the presence of LCPAs (Wenzl et al., 2008). Additional proteins contribute to specific steps in the process: 1) Cingulins, which are rich in tryptophan and tyrosine, localize to the girdle bands of diatom frustules; 2) Ammonium fluoride insoluble proteins (AFIM) are hypothesized to serve as pattern-forming base layers, and 3) silicalemma associated proteins (SAPs), including Silicanin-1, probably function as intermediaries between the

cytoskeleton and the SDV (Görlich et al., 2019; Kotzsch et al., 2017; Tesson et al., 2017).

   Silica polymerization in sponges is also biologically regulated and occurs both intracellularly and extracellularly. Intracellularly polymerization occurs within SDVs located in sclerocytes, the specialized silica-secreting cells. For spicules longer than ca. 250-300 µm, initial formation begins inside the sclerocyte but continues extracellularly, being transported out of the cell for further elongation, although the mechanisms underlying extracellular silicification remain poorly understood.

Some studies suggest that sclerocytes may either release exosome-like, silica-rich vesicles (silicasomes) or physically migrate along the growing spicule, depositing silica directly without the need for protein mediation (Müller et al., 2013; Schröder et al., 2007; Wang et al., 2012). More recently, specific enzymes involved in extracellular silica deposition have been identified in hexactinellid sponges (Shimizu et al., 2024), suggesting a more complex and regulated process than previously assumed. However, some evidence also indicates that not all large hexactinellid spicules are completed extracellularly, challenging the

assumption that size alone determines the transition from intra- to extracellular silicification (Mackie and Singla, 1997).

   Most research on the enzymatic control of silica polymerization in sponges is relatively recent. The first enzyme identified in this process, silicatein, was described by Shimizu et al. (1998). Silicateins are cathepsin-like proteins that catalyze the hydrolysis and condensation of silicic acid, facilitating controlled silica polymerization (Müller et al., 2007; Riesgo et al., 2015; Wang et al., 2012). These enzymes regulate polymerization through their surface hydroxyl groups and exhibit species-

specific variants that influence spicule morphology (Shimizu et al., 2015). Sponges also produce silicase, an enzyme that hydrolyzes silica under specific physiological conditions, potentially regulating silica solubility and remodeling during spicule growth (Ehrlich et al., 2010). Recent findings have identified additional proteins involved in silica polymerization, including silicatein-interacting proteins that modulate polymerization efficiency and structural integrity (Shimizu et al., 2024).



Furthermore, collagen-like and lectin-like proteins may play auxiliary roles in silica deposition by modulating silica-protein
interactions (Shimizu et al., 2024; Wang et al., 2012).

These biomineralization mechanisms evolved independently across the three classes of siliceous sponges. Demosponges
rely on silicateins for silica deposition, whereas hexactinellid sponges lack silicateins and instead use hexaxilin for intracellular
polymerization, along with glassin and perisilin for extracellular silica deposition that thickens spicules (Shimizu et al., 2015,
2024). Homoscleromorph sponges also display unique silica polymerization mechanisms, though the enzymes involved remain
unidentified. The presence of unrelated enzymes in the Class Demospongiae and Hexactinellida, along with the distinct
mechanisms observed in Homoscleromorpha, suggests that silicification has evolved multiple times independently in sponges
(Shimizu et al., 2024). This convergent evolution underscores the adaptive significance of silica biomineralization in marine
environments and highlights the diverse molecular strategies underpinning sponge silicification.

In plants, it is increasingly accepted that beyond the simple effect of monosilicic acid concentration beyond saturation
levels after water loss with transpiration, some cell wall polymers and proteins can initiate and control silicification (Zexer et
al., 2023). Essentially, polymers carrying hydroxyl groups, such as hemicellulose, callose or lignin, may stabilize silicic acid,
concentrate it, and favor its precipitation as amorphous silica. Beyond cell-wall type silica, it seems that proteins can also favor
silica deposits in cell lumens. For instance, the deposits of silica in the so-called silica cells found in many grass species are
thought to be controlled by a protein named Siliplant1 (Kumar et al., 2020b). These works have been pivotal to further
understanding plant silicification and its potential links to other silicifying organisms (Kumar et al., 2020a).

## 2.3. Environmental and ecological influences on silicification

While the regulated flow of silica ions and precursors is crucial for biomineralization, it is not the only factor determining this
process. Biomineralization is influenced by a complex interplay of environmental and ecological conditions. Although most
studies have focused on diatoms, similar mechanisms likely occur in other silicifying organisms. Silica deposition depends
directly on dSi availability, with its limitation resulting in reduced silicification (Brzezinski et al., 1990; McNair et al., 2018;
Shrestha et al., 2012). Lower temperatures have also been associated with increased silicification in diatoms (Durbin, 1977).
Additionally, under replete silica conditions, silicification tends to be inversely correlated with growth rate, suggesting that
other growth-limiting factors, such as the availability of iron ($Fe^{2+}/Fe^{3+}$) and zinc ($Zn^{2+}$), may indirectly regulate frustule
formation (Flynn and Martin-Jézéquel, 2000; Martin-Jézéquel et al., 2000). Light intensity further influences silicification in
a complex manner, with increased silicification observed under both low (15 µmol photons $m^{-2}$ $s^{-1}$) and high (300 µmol
photons $m^{-2}$ $s^{-1}$) light conditions (Petrucciani et al., 2023; Su et al., 2018; Xu et al., 2021). Other environmental factors, such
as salinity (Vrieling et al., 2007) and pH (Hervé et al., 2012), can also modulate silicification, emphasizing the intricate
interplay between environmental conditions and biomineralization.

Beyond its biochemical and environmental determinants, silicification plays a fundamental ecological role for silicifying
organisms contributing to their overall fitness. In diatoms, frustules act as protection against predators thanks to high
mechanical strength (Hamm et al., 2003). More heavily silicified diatoms are often rejected as food sources by copepods





(Ryderheim et al., 2022), and increased silicification has been observed in response to predator presence (Petrucciani et al., 2022a; Pondaven et al., 2007). This suggests that the trophic chain can influence biomineralization processes and affect Si bioavailability. Similarly, some sponge species allocate more resources to skeleton formation as a defense strategy against predation, thereby enhancing their chances of survival (Ferguson and Davis, 2008; Rohde and Schupp, 2011). In terrestrial plants, silica deposition enhances resistance to herbivory by making leaves more abrasive and harder to digest (Hartley and DeGabriel, 2016; Massey et al., 2006). However, studies on silica-based defenses in seagrasses are currently lacking, and it remains unclear whether these marine plants use similar strategies.

### 3. Diversity of Marine Silicifiers

Over the past decade, research has transformed our understanding of the marine Si cycle by revealing that multiple groups of organisms contribute to silica dynamics, challenging the long-standing paradigm that diatoms were the sole important silicifiers. Historically, diatoms have been considered the primary drivers of bSi production and recycling in the ocean (Nelson et al., 1995; Tréguer et al., 1995; Tréguer and De La Rocha, 2013). However, recent studies have highlighted the significant role of other groups, such as Rhizaria and sponges (e.g. Laget et al., 2024; Llopis Monferrer et al., 2020; Maldonado et al., 2019), while many additional taxa have been identified as interacting with Si (Baines et al., 2012; Churakova et al., 2023); **Fig. 2**). However, their contributions to Si cycling have yet to be quantified on regional and global scales. This expanding knowledge emphasizes the need to integrate multiple biological contributors into the Si biogeochemical cycle to better understand its functioning and enhance our ability to predict the effects of climate change and anthropogenic impacts (Tréguer et al., 2021).

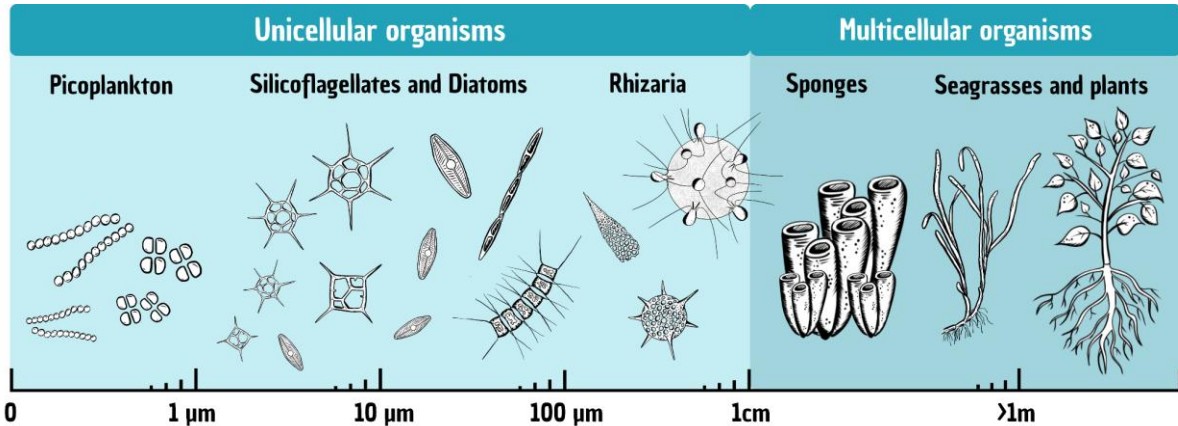

**Figure 2. Total size range of silicifiers, from the smallest unicellular organisms (picoplankton) up to multicellular terrestrial plants.**

The following paragraphs provide an overview of the main groups of silicifiers, ordered by their average size ranges.





## 3.1. Picoplankton


Picoplankton is a broad size category encompassing all planktonic or floating organisms measuring under 2 µm, though this is often extended to 3 µm due to the filter sizes used in field studies (Vaulot et al., 2008). The three main groups of picoplankton include heterotrophic bacteria, autotrophic bacteria (picocyanobacteria from the genera *Prochlorococcus* and *Synechococcus*), and autotrophic eukaryotes (picoeukaryotes), with picocyanobacteria and picoeukaryotes typically grouped together under the moniker picophytoplankton. The discovery of large abundances of heterotrophic bacteria in the oceans, larger than previously estimated, first put these tiny unicellular organisms on the map (Hobbie et al., 1977). Since then, picoplankton have been discovered to be the base of the microbial food web, and their significant contributions in terms of biomass and productivity make them influential players in global biogeochemical cycling (Azam et al., 1983; Falkowski et al., 2008; Le Quéré et al., 2005).


Picoplankton are distributed widely in oceans and freshwater environments, often described as ubiquitous or cosmopolitan. However, different picoplankton groups fill different environmental niches. For example, *Synechococcus* and picoeukaryotes prefer lower temperatures, while *Prochlorococcus* thrives in higher temperatures and lower latitudes (Buitenhuis et al., 2012; Flombaum et al., 2020; Visintini et al., 2021). *Prochlorococcus* abundances are notoriously high in oligotrophic (low nutrient) areas of the ocean, which can be explained by their competitive advantages (e.g. extremely small size, high diversity; Biller et al., 2015). Meanwhile, in nutrient-replete upwelling areas picoeukaryote growth is favoured (Li et al., 2023; Painting et al., 1993). Additional environmental factors that can affect picoplankton distribution patterns are light intensity and water mass events, as well as biological factors like the presence of predatory microplankton and viruses (Fuhrman, 1999; Otero-Ferrer et al., 2018; Sieradzki et al., 2019). Models predicting the impact of climate change on picoplankton suggest a global increase in picophytoplankton abundances, particularly in areas like the Arctic Ocean and Indian Ocean (Flombaum et al., 2020). In contrast, heterotrophic bacterial biomass in surface waters is expected to slightly decline, though this decrease is minor compared to the predicted reductions in microplankton and larger organisms (Heneghan et al., 2024; Kim et al., 2023). Overall, these findings indicate that future oceans will be dominated increasingly by picoplankton.




Picoplankton generally lack structural cellular components made of Si. However, some exceptions exist, such as *Triparma laevis* from the class Bolidophyceae, a close relative of diatoms that can measure less than 3 µm in size (Booth and Marchant, 1987; Kuwata et al., 2018; Yamada et al., 2014). Nevertheless, there is strong evidence that picoplankton interact with dSi in their environments regardless of whether or not they possess siliceous structures. Both extracellular and intracellular incorporation of dSi has been observed in heterotrophic bacteria. In some species of the bacterium *Bacillus*, dSi uptake occurs during early stationary phase and is used for spore formation (Hirota et al., 2010). Environments where dSi is saturated, particularly geothermal hot springs, are sites of extracellular silica polymerization in different bacterial species (Inagaki et al., 2003; Lalonde et al., 2005; Phoenix et al., 2000). In the marine environment, *Synechococcus* cells from the Sargasso Sea were found to harbour highly concentrated quantities of bSi (Baines et al., 2012). BSi accumulation and/or dSi uptake has been demonstrated by marine *Synechococcus* strains in both the natural environment and in laboratory experiments (Aguilera et al.,







2025; Krause et al., 2017; Ohnemus et al., 2016). The effect of dSi on picoplankton has generally been considered neutral, as *Synechococcus* growth rates do not change when cultured in a wide range of dSi concentrations, though a new study

demonstrates that high silicic acid concentrations can have negative effects on cell physiology (Ou et al., 2025). BSi accumulation has also been demonstrated by cultures of marine and brackish picoeukaryotes, including the smallest known marine eukaryote *Ostreococcus tauri* (Churakova et al., 2023). Curiously, in freshwater *Synechococcus* bSi accumulation was not observed, suggesting that *Synechococcus* strains in this environment lack the same mechanism for dSi uptake or do not utilize dSi in ways similar to marine or brackish *Synechococcus* (Aguilera et al., 2025). A mechanistic explanation for Si

uptake in these non-silicifying organisms can partially be explained by the SIT-Ls found in a limited number of bacterial and picocyanobacterial genomes (see **Section 2** and Marron et al., 2016). However, in the majority of species and strains studied, the mechanisms by which picoplankton uptake and utilize dSi need further investigation and more relevant methodologies for studying picoplankton need development.

### 3.2 Silicoflagellates

Silicoflagellates are marine planktonic organisms belonging to the class Dictyochophyceae (Silva, 1982), order Dictyochales (Haeckel, 1887), characterised by a distinctive star-shaped silica skeleton (mostly 20–100 µm). They are exclusively marine unicellular protists, mostly being considered photosynthetic, however, studies also show mixotrophic behaviour or non-photosynthetic bacteriovory (Eckford-Soper and Daugbjerg, 2016; Gerea et al., 2016; Sekiguchi et al., 2002). They have chromatophores for photosynthesis, but also possess pseudopodia, a typical feature of zooplankton, further complicating their

classification (Sheath and Wehr, 2003).

Almost all of their protoplasm is located inside the skeleton, which is composed of hollow tubular elements joined at triple junctions to create a tridimensional star-shaped structure constructed outside of the cytoplasm. They have a single flagellum situated at the anterior end of the cell, which extends outside the skeleton itself (Chang et al., 2017a). In culture, they have occasionally been observed to shed their skeletons, especially at high cell abundances, and continue either as swimming

flagellated cells or as unflagellated, amoeboid cells. These cells can divide without cytokinesis and develop into multinucleate organisms (Henriksen et al., 1993; Van Valkenburg and Norris, 1970). This naked stage has since been described in nature and has been found to occasionally occur in significant blooms (Jochem and Babenerd, 1989; Moestrup and Thomsen, 1990). Reproduction occurs through binary fission (Van Valkenburg and Norris, 1970). In rare cases, the formation of double skeletons has been observed with the two skeletons joined at the basal ring (McCartney et al., 2014). Though other modes of

reproduction have not been observed, the possibility of a sexual reproduction phase — similar to that of other phytoplanktonic organisms (e.g. diatoms) — cannot be ruled out. Silicoflagellate cell physiology remains largely understudied (Sancetta, 1990), partially due to challenges in maintaining them in culture. Despite the first successful culture in 1970 (Van Valkenburg and Norris, 1970), few cultures exist today, so individuals must be isolated from field samples for physiological experiments (Chang and Gall, 2016; Taguchi and Laws, 1985a).




Silicoflagellates first appeared in the Early Cretaceous (115 million years ago; McCartney et al., 2014) and showed great diversification throughout the Cenozoic (Anon, 1995; McCartney et al., 2018). Their distinct skeleton structure and morphology are the basis for taxonomic identification (Malinverno, 2010) and due to their high abundance in sediments, they are used in micropalaeontology as proxies for palaeotemperature reconstructions (see **Box 2** for further details). In the modern ocean, silicoflagellates are thought to be composed of four extant genera (*Dictyocha* (Ehrenberg, 1837)*, Octactism* (Schiller,

1925), *Vicicitus* (Chang et al., 2012),  and *Stephanocha* (Jordan and McCartney, 2015). However, as there is strong variability and plasticity in the skeleton (Dumitrica, 2014; McCartney et al., 2014; Van Valkenburg and Norris, 1970), in addition to the presence of the naked stage, this has led to considerable controversy regarding the number of species and genera (Chang et al., 2012, 2017a; Jordan and McCartney, 2015). The abundances of different silicoflagellate species are influenced by variations in temperature and salinity both in cultures (Henriksen et al., 1993; Van Valkenburg and Norris, 1970) and natural

environments (Hernández-Becerril and Bravo-Sierra, 2001; Ignatiades, 1970; Rigual-Hernández et al., 2016; Sancetta, 1990). However, the responses of individual species to these environmental factors are not always consistent (Sancetta, 1990). Additionally, Maliverno et al. (2010) demonstrated that the same species, *Stephanocha speculum*, can present different varieties linked to their precise ecological niches, with some varieties (*S. speculum* var. *monospicata*, var. *bispicata*, and var. *speculum)* being found in open ocean conditions, while *S. speculum* var. *coronata* is associated with sea ice cover.

Silicoflagellates are most abundant in nutrient-rich areas of the ocean at high latitudes (Fagerness, 1984; Henriksen et al., 1993; Onodera et al., 2016; Takahashi et al., 2009) and in upwelling systems (e.g. Hernández-Becerril and Bravo-Sierra, 2001; Murray and Schrader, 1983; Sancetta, 1990). Their carbon uptake rates have been shown to be comparable to other phototrophic phytoplankton (e.g. for *Dictyocha perlaevis;* Taguchi and Laws, 1985b), though there are no studies to date comparing different species of silicoflagellates or changes in environmental conditions. Despite occasional high

abundances, they are mostly considered insignificant contributors to primary production, planktic dynamics, and export fluxes. For example, in the Arctic Ocean they have been shown to contribute less than 1% of the flux of particulate organic carbon (Onodera et al., 2016). However, they are regionally important prey for different copepod and krill species (Nakagawa et al., 2002; Passmore et al., 2006; Pasternak and Schnack-Schiel, 2001) as well as gelatinous grazers (Hiebert et al., 2025). Some species of silicoflagellates (e.g. *Dictyocha* spp.) have additional ecological significance through their role as harmful algae

bloom (HAB) species with significant ichthyotoxicity (Esenkulova et al., 2020; Henriksen et al., 1993).

 While the ecological relevance of silicoflagellates has been recognised, their mechanisms of silicic acid uptake remain unexplored, and their contribution to the Si and carbon cycles has yet to be quantified. While studies focused on quantifying the regional contribution of silicoflagellates to Si export (Takahashi, 1989), their role in global Si cycling is entirely unconstrained (Tréguer et al., 2021).

**3.3. Diatoms**

Diatoms are a greatly diversified group of eukaryotic phytoplankton (Bacillariophyceae) belonging to the supergroup Stramenopiles (Bowler et al., 2010). They were first observed in 1703 (Round et al., 1990) and, since then, have attracted





scientific interest across multiple disciplines, from taxonomists, oceanographers, and geologists, to more recently engineers
and chemists, establishing them as the by far best-studied group of marine silicifiers. These photoautotrophic unicellular

microorganisms are ubiquitous and can be found in all aquatic environments, including in sea ice and in soils, lichens, biofilms
and airborne, denoting their extensive adaptive capabilities (Lohman, 1960).

Diatoms are characterised by their ability to exploit silicic acid to build a siliceous shell called a frustule. Diatom taxonomy
is based on frustule morphological traits and categorizes the species in two groups based on cellular symmetry: centric diatoms,
with a radial symmetry, and pennate diatoms, with an elongated and bilateral symmetry (Round et al., 1990). Diatoms are

thought to be secondary endosymbionts (Gentil et al., 2017; Morozov and Galachyants, 2019), meaning that their precursor
was a eukaryote that phagocytized another eukaryote (which became an organelle), resulting in chloroplasts surrounded by 4
membranes. The earliest known fossil record of diatoms is still widely debated (Brylka et al., 2023), but molecular clock and
sedimentary evidence suggest an earlier origin, near the Triassic-Jurassic boundary (Nakov et al., 2018), when a greater
availability of niches occurred. The gap in fossil records during this period has been hypothesized to relate to non-silicified or

lightly silicified diatoms (Armbrust, 2009).

Diatoms are key primary producers in nowadays oceans and their ability to adapt to different environmental conditions
allows them to outcompete other phytoplanktonic groups during blooms. They are highly efficient primary producers
constituting 40% of marine primary production or 20-25% of global Earth's primary production, despite representing only 1%
of Earth's photosynthetic organic biomass (Falkowski et al., 2004; Field et al., 1998). The frustules improve diatom fitness

compared to other microorganisms (Martin-Jézéquel et al., 2000) including but not limited to providing mechanical protection,
allowing hydrodynamic control of particle diffusion and advection, or increasing photosynthetic efficiency.

Nutrient uptake in diatoms is facilitated by their large vacuoles, which store essential elements like nitrogen and Si. This
storage capacity supports rapid growth during favourable conditions, contributing to their ecological success (Behrenfeld et
al., 2021). Diatoms reproduce asexually through mitotic division, which progressively reduces cell size due to the constraints

of their rigid silica frustules. During each division, the daughter cells inherit one parental valve and form a new, smaller valve
within it, leading to a gradual decrease in cell dimensions over successive generations (Jewson, 1997). When cells approach a
critical minimum size threshold, sexual reproduction is triggered to restore maximal cell dimensions. This process involves
the formation of auxospores, specialized zygotic cells that expand and produce the initial large vegetative cells for the next
cycle (Chepurnov et al., 2002). Auxospore formation is influenced by environmental factors such as light, nutrient availability,

and cell density, as well as endogenous chemical signals (Assmy et al., 2006; Mouget et al., 2009). The interplay between
asexual size reduction and sexual size restoration ensures diatom populations maintain their ecological and evolutionary
success. This unique life cycle strategy contributes to their adaptability and widespread distribution in aquatic ecosystems.

Their ecological success marks diatoms as important contributors of biogeochemical cycles. Globally, diatoms produce
approximately 255 Tmol Si yr$^{-1}$ of bSi, with regional variations influenced by nutrient availability and oceanic regimes

(Tréguer et al., 2021). Coastal areas exhibit silica production rates 3–12 times higher than the global average, while
oligotrophic gyres show rates 2–4 times lower (Brzezinski et al., 1998). Nutrient limitation, particularly of Si, iron, and



nitrogen, is a key driver of diatom growth and silica production. In oligotrophic gyres, low dSi concentrations (0.9–3.0 µM) limit silica production, despite rapid diatom growth rates. In nutrient-rich zones, diatom blooms significantly enhance silica and carbon export, with Si:C ratios ranging from 0.09 to 0.15 depending on species and environmental conditions (Brzezinski, 1985). Their regionally high abundance and ubiquitous distribution make them a key source of organic carbon for higher trophic levels, including zooplankton and fish larvae. Moreover, diatoms are an important vector for carbon export (~20-40%; Jin et al., 2006) due to their ability to form aggregates and their dense frustule, which improves sinking. As they sink, they rapidly transport organic matter to deeper ocean layers, and eventually are sequestered in sediments over geological time scales thanks to their siliceous frustules that enhance their preservation within the sedimentary records. Due to the well-defined ecological niches occupied by individual diatom species, their fossil assemblages serve as a reliable proxy for reconstructing past environmental conditions and tracking climate evolution (see **Box 2** and e.g. Crosta et al., 2021; Torricella et al., 2025).

**3.4 Siliceous Rhizaria**

Rhizaria, a diverse 'supergroup' of unicellular eukaryotes, represents a fascinating and largely unexplored realm of biodiversity (Burki and Keeling, 2014). This supergroup is subdivided in three phyla: Endomyxa, Cercozoa, and Retaria. A key unifying morphological feature within these phyla is the production of very fine pseudopodia, often numerous, and typically supported by microtubules called filopodia (Adl, 2024). The major taxa of siliceous Rhizaria are represented by organisms belonging to the Cercozoa and Retaria phyla. Among Cercozoans with skeletons, those belonging to the Thecofilosaria, especially the Phaeodaria, possess skeletons that take the form of spicules or bars, which vary in size and shape. Although Phaeodaria were traditionally classified as Radiolaria due to morphological similarities, molecular phylogenetic analyses have demonstrated that they are not closely related to Radiolaria and are instead affiliated with the Cercozoa (Nikolaev et al., 2004). The phylum Retaria includes organisms with skeletons of various compositions, and is divided into Foraminifera and Radiolaria. The skeletons of Foraminifera are primarily made of calcium carbonate, while Radiolaria exhibit skeletons of various nature, including strontium sulfate (exclusive to the Acantharia class) or opaline silica (found in the Polycystine and Sticholonche classes).

Phaeodaria and Radiolaria, both characterized by an intricate siliceous skeleton, exhibit a remarkable range of lifestyles (solitary and colonial) and trophic diversity (heterotrophic or mixotrophic with photosynthetic symbionts). They also present great morphological variability, with sizes spanning from a few tens of micrometres to several millimetres, gigantic sizes for protists. A key structural distinction lies in their skeletons. While Radiolaria skeletons are composed of robust, solid amorphous silica, Phaeodaria skeletons are hollow (Takahashi, 1983). Among the Polycystine Radiolaria, Spumellaria and Nassellaria are the most extensively studied groups, largely due to their robust skeletons and continuous fossil record dating back to the early Cambrian (De Wever et al., 2001). Spumellaria are characterized by concentric, generally spherical skeletons and were the first to appear in the Cambrian period. In contrast, Nassellaria display bilateral symmetry and possess conical skeletal structures with a limited number of spicules, first emerging in the Early Triassic (Anderson, 2001). Nassellaria are also known





for their ability to form colonies, such as those seen in Collodaria, or large solitary skeletons, such as those
of Orodaria (Nakamura et al., 2021).

Siliceous Rhizaria are widely distributed across the world's oceans, from tropical to polar regions, and are especially abundant in open-ocean environments. These protists are commonly found in the upper layers of the ocean, where sunlight supports the photosynthetic symbionts hosted by many Rhizaria species (Decelle et al., 2021). However, they are also well represented in deeper zones, including the mesopelagic (Biard and Ohman, 2020). Global assessments indicate that Rhizaria
play a substantial role in zooplankton abundance, contributing up to 33% (Biard et al., 2016). More recent studies, also using advanced imaging techniques to quantify large Rhizaria, indicate that they represent up to 1.7% of mesozooplankton carbon biomass within the upper 500 m of the water column. Interestingly, Rhizaria biomass, particularly that of Phaeodaria, is more than twice as high in the mesopelagic than in the epipelagic layer (Laget et al., 2024). In marine ecosystems, Rhizaria have also been shown to dominate the eukaryotic community captured in sediment trap samples collected below the euphotic zone,
as revealed by metabarcoding analyses (Gutierrez-Rodriguez et al., 2019; Preston et al., 2020).

The reproductive biology of Rhizaria remains poorly understood, primarily due to the challenges of maintaining most species under laboratory conditions. Although various life stages and lineage-specific cellular innovations have been observed in natural environments, their functional roles, as well as their physiological and ecological significance, remain largely uncharacterized (Rizos et al., 2024). Evidence suggests that both sexual and asexual reproduction occur across different
Rhizaria lineages. Sexual reproduction involves the production and release of haploid gametes, which fuse to form a zygote with genetic material from both parent organisms. Asexual reproduction occurs by cell division during mitosis, resulting in two identical organisms. In colonial forms, asexual growth may occur through fission of the central capsules, contributing to colony expansion or fragmentation into multiple new colonies. In several species, reproduction involves the release of numerous flagellated swarmer cells that are considered to be gametes, produced via multiple nuclear divisions within the parent
cell. These swarmers are released simultaneously and are presumed to fuse into zygotes, although the exact mechanisms of gamete fusion and early development remain poorly understood (Suzuki and Not, 2015). While the early ontogenetic stages are still not fully documented, the process of skeletal formation during growth has been extensively studied in several species under laboratory conditions (Anderson, 2001).

The widespread distribution of siliceous Rhizaria across marine ecosystems underpins their ecological importance in both
food web dynamics and biogeochemical cycles. As basal components of planktonic communities, they serve as essential food sources and can act as vectors for energy transfer to higher trophic levels. Many Rhizaria host photosynthetic symbionts, enhancing primary production in oligotrophic waters (e.g. Llopis Monferrer et al., 2025). For instance, Collodaria are particularly abundant in nutrient-poor, tropical epipelagic zones and contribute significantly to zooplankton biomass (Faure et al., 2019). In deeper waters, Rhizaria play an important role in transforming sinking particles and serve as a key source of
organic carbon in the mesopelagic zone (Laget et al., 2024; Lampitt et al., 2023). Their ecological significance extends to the Si cycle, as many Rhizaria species take up silicic acid from seawater to build their intricate skeleton (Llopis Monferrer et al., 2020) contributing to approximately 20% of the bSi production in the global ocean (Tréguer et al., 2021). Through both



biological uptake and export, Rhizaria facilitate the vertical transport of silica and carbon to the deep ocean, reinforcing their role in long-term elemental cycling (Biard et al., 2018; Stukel et al., 2018). The opaline silica skeletons they produce also

contribute to a rich and continuous fossil record since the Cambrian, more than 500 million years ago (**Box 2**; De Wever et al., 2001).

## 3.5 Sponges

Sponges belong to the phylum Porifera, which comprises approximately 9,750 species classified into four distinct classes: Demospongiae, Calcarea, Hexactinellida, and Homoscleromorpha (Hooper and Van Soest, 2002; de Voogd et al.,

2025). The largest and most diverse class, Demospongiae, accounts for nearly 84% of all known sponge species. The remaining species are distributed among the classes Calcarea (8%), Hexactinellida (7%), and Homoscleromorpha (1%; Van Soest et al., 2012). Sponges have been part of marine fauna since the late Precambrian (890-580 Mya) and are among the earliest known multicellular organisms on Earth (Wörheide et al., 2012). Since the Early Cambrian, many sponges have evolved siliceous skeletons, a trait present in over 80% of extant species (Neuweiler et al., 2014). These sessile filter feeders inhabit a vast range

of marine habitats, from intertidal zones to the deep sea, across all latitudes (Maldonado et al., 2017; Van Soest et al., 2012). While some species have broad distributions, others are highly specialized, restricted to specific substrates or environmental conditions such as deep-sea sponge reefs, coral-associated habitats, or high-silica environments.

Due to their ubiquity and substantial biomass, sponges are increasingly recognized as key functional components of ocean ecosystems (Bell, 2008; Bell et al., 2023; Folkers and Rombouts, 2020). Their complex body structures provide habitat for a

diverse array of sessile and mobile organisms, making sponge-dominant communities biodiversity hotspots (Beazley et al., 2013; Klitgaard, 1995). Additionally, sponges contribute significantly to biogeochemical cycling by filtering and recycling dissolved organic and inorganic matter and nutrients, thereby influencing the fluxes of essential elements such as carbon, nitrogen, phosphorus, and Si (Busch et al., 2022; De Goeij et al., 2013; Maldonado et al., 2012).

Sponges reproduce both sexually and asexually. In sexual reproduction, gametes are released into the water column for

external fertilization, although some species exhibit internal fertilization (Maldonado and Riesgo, 2008). Asexual reproduction occurs through budding, fragmentation, or gemmule formation, contributing to their resilience in dynamic environments (Battershill and Bergquist, 1990). Sponges exhibit a biphasic life cycle, alternating between a pelagic larval stage and a sessile adult stage. Larvae are typically short-lived, ranging from hours to a few days, and rely on passive dispersal by ocean currents (Maldonado and Abdul Wahab, 2025). Upon settlement, larvae undergo metamorphosis into juvenile sponges, attaching to a

substrate where they develop into mature, filter-feeding adults. Sponges generally live for several years to decades, with some species known to survive even for centuries (Leys and Lauzon, 1998; McGrath et al., 2018; McMurray et al., 2008).

For decades, the role of sponges in Si cycling was considered negligible. However, recent research has demonstrated their significant contribution at both regional and global scales (López-Acosta et al., 2022; Maldonado et al., 2019, 2021). Siliceous sponges actively assimilate large amounts of dSi, transforming it into bSi for skeletal formation. Upon their death, this silica

can either dissolve back into the water column, sustaining other silicifiers, or be buried in sediments, influencing long-term





silica sequestration. This process is particularly relevant in deep-sea sponge grounds and continental shelf ecosystems, where sponges are particularly abundant and act as major reservoirs facilitating benthic-pelagic coupling of the Si cycle (Tréguer et al., 2021). Furthermore, sponge-mediated Si cycling interacts with other major biogeochemical cycles such as those of carbon and nitrogen, as sponge excretion and microbial symbionts further contribute to matter and nutrient transformations (Busch et 530 al., 2022; Maldonado et al., 2012).

## 3.6 Plants

In the last two decades, an important number of functions has been assigned to leaf silicification (Cooke and Leishman, 2016; Johnson et al., 2024), including for wetland and aquatic plants (Schoelynck et al., 2012; Schoelynck and Struyf, 2016). Concentrations of Si in wetland and aquatic plant species are highly variable, and can sometimes reach very high values 535 (Schoelynck and Struyf, 2016). For instance, concentrations of Si in yellow water-lily (*Nuphar lutea*) can go up to 8% in some individuals. Interestingly, species that are emergent (only basal stem parts in the water) and submerged tend to accumulate more Si compared with species that mostly have floating organs (Schoelynck and Struyf, 2016). Such difference has been interpreted as resulting from the structural role of silicification (Schoelynck and Struyf, 2016). In emergent and submerged species, silicification could be used to increase strength in order to overcome different physical stresses (gravity, horizontal 540 force, etc.). In floating species, on the contrary, silicification would be less important because they are less impacted by hydrodynamics forces (Schoelynck and Struyf, 2016). Such link between silicification and physical stress is supported by evidence of trade-offs with others C-based structural components, such as cellulose and lignin (Schoelynck et al., 2010), and by links between wind speed and Si accumulation in terrestrial herbaceous species (Song et al., 2020). Beyond the structural role of Si in wetland and aquatic species, Si is also used for better resisting herbivory. For instance, the bSi concentration of 545 different *N. lutea* individuals is negatively correlated with the damages caused by *Galerucella nymphaeae* (Schoelynck and Struyf, 2016). Whether silicification is an adaptation or exaptation to grazing is, however, still debated (de Tombeur et al., 2023b). While trait-based approaches have recently advanced our understanding of silicification strategies in plants, applying similar frameworks to other marine silicifiers such as diatoms remains challenging due to fundamental differences in morphology and silica utilization patterns (see **Box 3**).

In coastal systems, seagrass meadows provide vital ecological services, including nutrient cycling and climate regulation, and may influence Si dynamics in estuarine and marine environments. Despite their ecological importance and proximity to riverine Si inputs, the role of seagrasses in the Si cycle remains understudied. While the elemental composition of seagrass leaves has been widely analysed for macronutrients like nitrogen and phosphorus, only a few studies have quantified Si concentration in seagrass (see Herman et al., 1996; Roth et al., 2025; Vonk et al., 2018), leaving an important gap in 555 understanding their role in biogeochemical Si cycling. The mechanisms of Si accumulation in seagrasses are not yet fully understood, but recent research in *Zostera marina* has identified phytoliths in its roots, stems, and leaves (Rong et al., 2024) and suggests a functional dSi uptake pathway involving the Slp1 protein and vesicular transport (Kumar et al., 2020b; Nawaz et al., 2020). By measuring bSi in four families of seagrass species, Vonk et al. (2018) showed that *Z. marina* tends to





accumulate more silica than other seagrasses, though still less than freshwater vegetation. This species displays a relatively
low bSi content but may still play a significant role in coastal Si cycling, especially since some of its Si resists dissolution and
could buffer the transport of dSi to the ocean (Roth et al., 2025).

## 4. Impact of living organisms on marine silicon cycling

Understanding the diversity, biology, and ecological roles of silicifying organisms is essential to contextualize their broader
influence on Earth system processes. These organisms not only construct siliceous structures but also actively shape the marine
and terrestrial Si cycles through uptake, recycling, and burial of bSi. Their evolutionary emergence and ecological expansion
have fundamentally altered the distribution and availability of dSi in aquatic environments over geological timescales. In the
following section, we examine how the biogeochemical cycle of Si has evolved from abiotic to biologically mediated control,
highlighting the role of silicifying taxa in transforming Si fluxes, sedimentation patterns, and ocean chemistry across Earth's
history and into the modern era.

### 4.1. Evolution of the silicon cycle across geologic time

Researchers began trying to better understand the evolution of the oceanic Si cycle just over three and a half decades ago
(Maliva et al., 1989; Siever, 1991, 1992), with considerable effort since then focused on standing stocks and how the current
cycle evolved (Conley et al., 2017; Trower et al., 2021). The modern oceanic Si cycle is relatively well studied (Tréguer et al.,
2021), with Si mainly occurring in the marine environment in the form of dSi. The amount of dSi available in the ocean is a
balance between inputs (e.g. rivers, aeolian, submarine groundwater, glaciers, and hydrothermal activity), biological
circulation via silicifiers (e.g. diatoms, sponges, and Rhizaria), and removal (e.g. burial, biogenic uptake, and secondary
formation; see **Fig. 3**). Overall, modern oceanic dSi concentrations are low, averaging less than 10 μM at the surface and
around 70 μM in the deep ocean (Gouretski and Koltermann, 2004; Tréguer et al., 1995), primarily regulated by biological
activity, but this has not always been the case. In the absence of silicifiers, the oceans must have had higher dSi concentrations
in the Precambrian compared to the Phanerozoic (Maliva et al., 2005). Understanding the evolution of silicifiers is therefore
crucial to understanding the geologic history of the oceanic Si cycle.





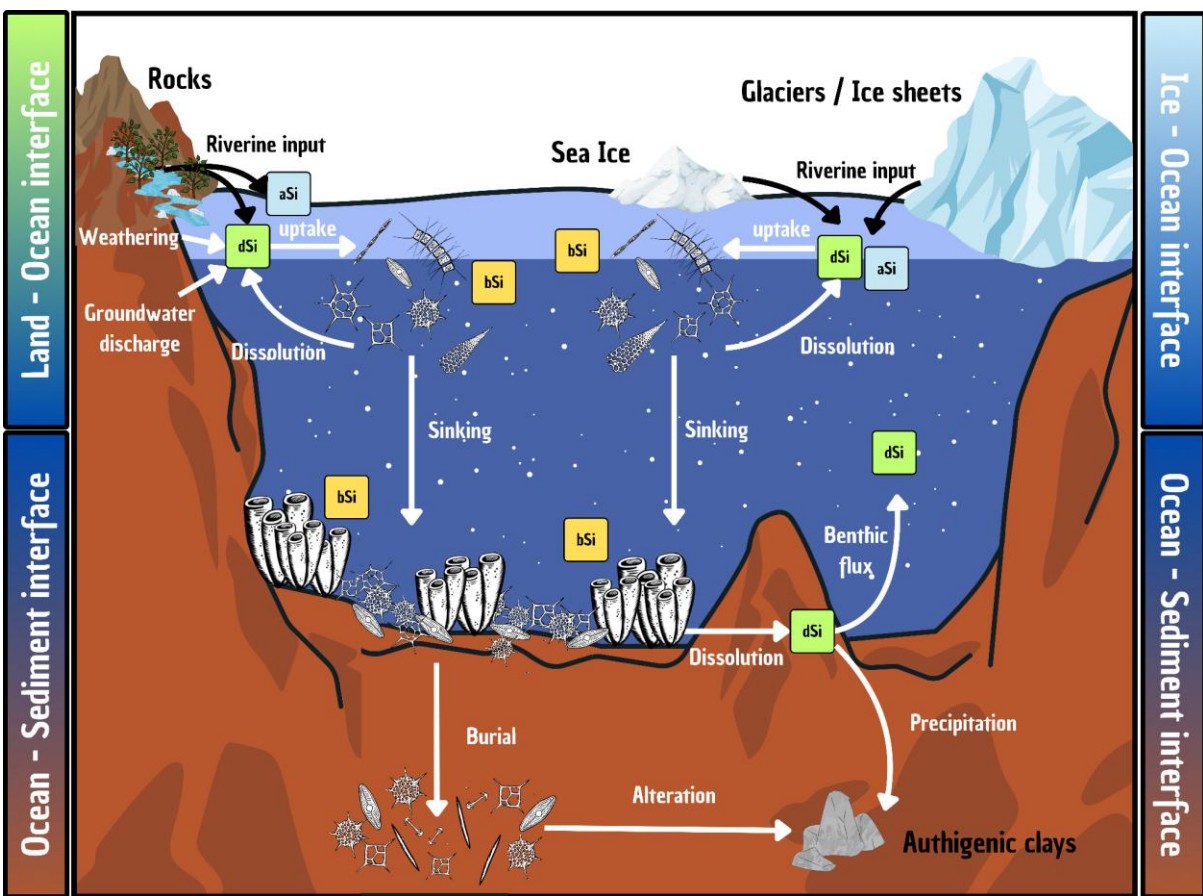

**Figure 3. Schematic representation of the Si cycle in the modern global ocean, illustrating key processes occurring at the interface**
**between the ocean and other environments (e.g. sediments, land, ice). Particular attention is given to the transformations and**
**transfers of silicon in both dissolved (dSi; green boxes) and particulate forms (e.g. biogenic silica, bSi - yellow boxes - and amorphous**
**silica, aSi - blue boxes) as it moves between the surface and deep ocean. Details are given in section 4.3 and quantification of the**
**fluxes are presented in Table 1.**

The Precambrian era (>540 Ma) covers most of the Earth's history with little to no complex life forms in the oceans and is characterized by its widespread abiotic silica-rich chert formations. Precambrian cherts are chemically precipitated sedimentary rocks that formed in peritidal and shallow shelf environments mainly during the Proterozoic (2500 – 540 Ma) (Perry and Lefticariu, 2007). Due to continental weathering, volcanism, and hydrothermal activity, the oceans were saturated with respect to dSi, and the Si cycle was mainly controlled by the solubility of mineral phases precipitating out of the seawater (Siever, 1991, 1992). For this reason, there is one main difference between the Precambrian and the Phanerozoic (<540 Ma), bSi precipitation. Silicifiers had not yet fully evolved in an abundance to efficiently remove dSi from the oceans and influence the Si cycle as they do today. Pinpointing the exact evolutionary timeline of dSi utilizing organisms is tricky, as fossil evidence for early biogenic silicification is viewed as controversial (Chang et al., 2017b; Porter and Knoll, 2000; Sperling et al., 2010). There are some Neoproterozoic (1000 – 540 Ma) protists which had scales that were probably siliceous (Morais et al., 2017),





and Ediacaran (635 – 540 Ma) sponges with siliceous spicules (Chang et al., 2019; Chen et al., 2023), but the full extent of their influence on the Si cycle is still unknown. The appearance of biogenic tools used during silicification challenges the idea that dSi in the Precambrian oceans was only controlled by inorganic reactions, and suggests that oceanic dSi could have been influenced by biology before the Phanerozoic began. The Precambrian Si cycle may therefore not only be abiotically controlled by inorganic reactions, volcanism, hydrothermal activity, and reverse weathering (Isson and Planavsky, 2018; Jurkowska and

Świerczewska-Gładysz, 2024; Maliva et al., 1989, 2005), but potentially the appearance of organisms capable of using Si (Conley et al., 2017; Marron et al., 2016). While the exact timeline and quantity is widely debated in the literature (e.g. Conley et al., 2017; Grenne and Slack, 2003; Maliva et al., 2005; Trower et al., 2021), following the Precambrian, a decrease in oceanic dSi concentrations (~500 – 1000 μM) occurred with the widespread evolution of large-scale skeletal marine silicification and the appearance and radiation of siliceous sponges.

The expansion of silica biomineralization via sponges at the beginning of the Paleozoic (540 – 420 Ma) (Chang et al., 2019) lowered dSi concentrations below saturation and changed the oceanic Si cycle to a biologically dominated system, which we continue to see today (Tréguer et al., 2021; Trower et al., 2021). Fossil evidence confirms this biological switch as it was observed that, as sponge populations began to increase, the thick layers of shallow abiotic silica-rich chert became less common and were replaced by deep ocean patchy hydrothermally influenced biogenic chert formations (Chang et al., 2019). Just after

the appearance of sponges, a siliceous zooplankton group, Radiolaria, emerged and quickly expanded worldwide during the Ordovician (485 Ma) (Fortey and Holdsworth, 1971; Kidder and Tomescu, 2016), resulting in an additional parallel drop in oceanic dSi concentrations. From there, sponges and radiolarians regulated dSi in the oceans for several million years via biomineralization and the uptake and burial of bSi.

The Mesozoic Era (250 – 145 Ma) saw a massive change in the oceanic ecosystem, with a significant decrease in dSi

concentrations. It is believed that the Mesozoic Si cycle was closer to modern day values due to the radiation of diatoms (Harper and Knoll, 1975; Maliva et al., 1989; Siever, 1991). The exact timeline of this major dSi drawdown is largely questioned (Conley et al., 2017; Trower et al., 2021; Yager et al., 2025), as the evolution and worldwide radiation of diatoms is still widely debated (Bryłka et al., 2023). Molecular clocks suggest that diatoms originated 200 million years ago near the Triassic-Jurassic boundary (Nakov et al., 2018), but the first widespread fossil deposits were found in the Lower Cretaceous

(121 Ma) (Bryłka et al., 2024), causing uncertainty in when their abundance took control of the global oceanic Si cycle.

While the major players remain the same, the current oceans differ from those of the Mesozoic in many aspects including sea level, tectonic seafloor spreading and a lack of epicontinental seas. Our present-day marine Si cycle, as presented in the next section, is likely the result of not only biological evolution but also tectonic and palaeoenvironmental conditions established in the Eocene (33.9 Ma), with low dSi concentrations and biological control of uptake, circulation and burial

(Jurkowska and Świerczewska-Gładysz, 2024).



## 4.2 The modern ocean Si cycle and its fluxes

### 4.2.1 Biogenic silica production of different silicifiers

Silicic acid levels in modern oceans are largely regulated by diatoms, with the biological activities of silicifiers playing a key role in shaping the marine geochemical cycle of Si. In the modern ocean, pelagic diatoms consume Si more efficiently and

produce bSi at a rate of $255\pm52$ Tmol-Si yr$^{-1}$ (Tréguer et al., 2021), which is higher than the annual production rate of picocyanobacteria (<20 Tmol-Si yr$^{-1}$, Brzezinski et al., 2017; Krause et al., 2017), rhizarians (2−58 Tmol-Si yr$^{-1}$, Llopis Monferrer et al., 2020), siliceous sponges ($6.2\pm5.9$ Tmol-Si yr$^{-1}$, Tréguer et al., 2021), benthic diatoms ($6.1\pm0.7$ Tmol-Si yr$^{-1}$ Tréguer et al., 2021), and macroalgae (0.27−1.33 Tmol-Si yr$^{-1}$, Yacano et al., 2021). As the bSi production of non-diatom silicifiers is minor and falls within margin of error for pelagic diatom production, Tréguer et al. (2021) set the preferred estimate

for global pelagic bSi production at $255\pm52$ Tmol-Si yr$^{-1}$.

Although pelagic diatoms are the primary producers of bSi in the global ocean, the contributions of other silicifiers to the marine bSi production are also important at both regional and global scales when considering their role in the marine food web and the export of carbon to depth via the biological Si pump. For instance, the accumulation of Si by picocyanobacteria accounts for around 10% of the bSi produced in the mid-latitude ocean gyres and oligotrophic oceans (Baines et al., 2012;

Krause et al., 2017; Wei et al., 2023). A recent estimate based on the Si:C ratio and picocyanobacteria cell abundances found that Si accumulation by picocyanobacteria may account for 12% of the global Si inventory and 45% of the global annual bSi production (Wei et al., 2023). However, it should be noted that Wei et al. (2023) applied the Si:C ratio of *Synechococcus* to *Prochlorococcus* (the most abundant species) in nutrient-depleted waters, where competition for dSi by diatoms is undermined, to the global ocean, which likely overestimates the role of picocyanobacteria in the marine Si cycle.

Benthic diatoms are another important group of silicifiers living in the coastal photic zone (Cahoon, 1999). The first estimate of global benthic diatom production was calculated based on: 1) a normalised production rate of benthic diatoms (1 mol-Si m$^{-2}$ yr$^{-1}$ applied to the global coastal photic zone, yielding 6.8 Tmol-Si yr$^{-1}$) and 2) global benthic microalgal primary production and the Si:C ratio of the diatom cells (i.e. (0.13 mol-Si mol-C$^{-1}$ × 514 × $10^{12}$ g-C yr$^{-1}$)/12 g-C mol$^{-1}$ = 5.4 Tmol-Si yr$^{-1}$; see Tréguer et al., 2021).

Due to their long lifespans and low dissolution rate, siliceous sponges store a considerable amount of Si within their skeletons (Chu et al., 2011; López-Acosta et al., 2022; Maldonado et al., 2019). They are widely distributed in the world's oceans and comprise a diverse range of species (Van Soest et al., 2012). Dense sponge reefs have been found in the Strait of Georgia (Chu et al., 2011), while sponge spicules have been found in mats exceeding 1 m in thickness in the Antarctic (Gutt et al., 2013). However, the quantification of the sponge bSi production on the global scale remains highly uncertain (DeMaster,

2019; Tréguer et al., 2021) because of the large variations in Si consumption among sponge species at different growth stages and in different regions (López-Acosta et al., 2016; López-Acosta et al., 2018; Maldonado et al., 2011), and the poorly constrained standing crop of siliceous sponges on the seafloor. Thus, Tréguer et al. (2021) estimated the sponge bSi production



by assuming the deposition rate of sponge bSi was equal to the sponge production rate, a method that does not require information on the sponge standing crop.

Furthermore, marine macroalgae and seagrasses also contain Si in their biomass, therefore, they may contribute significantly to bSi production in the global neritic zone. By analysing the Si content of various macroalgae genera, Yacano et al. (2021) found large variability in the Si content (ranging from 0.13% to 39.4% by dry weight). By scaling the global net primary production of macroalgae (80−210 Tmol-C yr$^{-1}$, see Raven, 2018) by the mean C:Si ratio of macroalgae (295.6) and the median C:Si ratio of macroalgae (157.4), Yacano et al. (2021) estimated the global macroalgae bSi production to be

0.27−1.33 Tmol-Si yr$^{-1}$. This is similar in magnitude to the global bSi production of sponges and benthic diatoms.

The area of seagrasses in the global ocean was estimated using field survey data (160387−266562 km$^2$; McKenzie et al., 2020) and modelling results (917169 km$^2$; Gouvêa et al., 2024), yielding large variations in the estimated seagrass meadow area. Based on the previously reported global seagrass shoot production rate (0.34 g DW m$^{-2}$ d$^{-1}$; Strydom et al., 2023), the Si content of seagrass leaves (0.082% DW; Vonk et al., 2018), and the Si reservoir in seagrass meadows (0.18 g-Si m$^{-2}$; Roth et

al., 2025) of the North Atlantic, we made a conservative estimation of the annual global seagrass bSi production (0.58−3.33×10$^{-3}$ Tmol-Si yr$^{-1}$) and the seagrass bSi reservoir (0.33−1.86×10$^{-3}$ Tmol-Si). This represents a lower limit of the global seagrass bSi production due to the low seagrass Si content (Vonk et al., 2018) used in the calculation and the low production rate in the North Atlantic compared to other regions such as the temperate North Pacific and the temperate Southern Hemisphere (Strydom et al., 2023). Previous studies had not considered the production of bSi by marine macroalgae and

seagrasses (DeMaster, 2019; Tréguer et al., 2021; Tréguer and De La Rocha, 2013). However, these marine organisms may produce an important amount of bSi in the neritic zone (Yacano et al., 2021; and this study). Further research is necessary to understand the bSi production of these silicifiers at different seasonal and regional scales.

### 4.2.2 Fluxes and budget of Si in the modern ocean

Silicon, which is ultimately derived from the weathering of the Earth's crust, is transported to the ocean via various pathways

(Tréguer and De La Rocha, 2013; **Fig. 3**). It is then removed from the ocean through the burial of bSi (Tréguer et al., 2021) and the formation of authigenic clays (Aller and Wehrmann, 2025; Rahman et al., 2017). Accurately quantifying the different fluxes of the marine Si budget is important for understanding how it evolves in response to global warming and ocean acidification, and other anthropogenic impacts (e.g. river damming and changes in land use).

The first Si budget was evaluated by Calvert (1968) and considered river and seafloor basalt weathering to be major marine

Si sources. The burial of bSi on the seafloor was identified as the only ocean Si sink (**Table 1**). However, further research has identified several additional significant sources of Si, including hydrothermal activity (DeMaster, 1981), aeolian input (Tréguer et al., 1995), submarine groundwater discharge (Tréguer and De La Rocha, 2013) and glacial meltwater in the polar regions (Tréguer et al., 2021) and the release of Si from sandy beaches (Aparicio et al., 2025). Several important sinks were also identified, including siliceous sponges (Maldonado et al., 2019; Tréguer and De La Rocha, 2013) and authigenic clay formation

(Michalopoulos and Aller, 1995; Tréguer et al., 2021; Tréguer and De La Rocha, 2013) (see **Table 1** and Section 4.3.3).





**Table 1. Si sources and sinks of the global ocean accessed during the past 60 years. The unit of the flux is expressed as Tmol-Si yr$^{-1}$ ($10^{12}$ mol-Si yr$^{-1}$). The numbers marked with "*" represent values calculated in this study, and values in parentheses are fractions of the total Si source or sink.**

| Marine Si Sources | Calvert (1968) | DeMaster (1981) | Tréguer et al. (1995) | DeMaster (2002) | Tréguer and De La Rocha (2013) | Frings et al. (2016) | DeMaster (2019) | Tréguer et al. (2021) | Aparicio et al. (2025) |
|---|---|---|---|---|---|---|---|---|---|
| River | 7.2 (99%) | 5.6 (90%) | 5.0±1.1 (82%) | 5.6 (84%) | 7.3±2.0 (67%) | 6.33±0.36 (62%) | 6.3±0.4 (53-81%) | 8.1±2.0 (55%) | 8.1±2.8 (38%) |
| Riverine amorphous Si | - | - | - | - | | 1.90±1.00 (19%) | 0.2±0.5 (2-3%) | | |
| Hydrothermal sources | - | 0.6 (10%) | 0.2±0.1 (3%) | 0.6 (9%) | 0.6±0.4 (6%) | 0.60±0.40 (6%) | 0.4±0.2 (3-5%) | 1.7±0.8 (11%) | 1.7±0.9 (8%) |
| Basalt weathering | 0.1 (1%) | - | 0.4±0.3 (7%) | | 1.9±1.7 (17%) | 0.40±0.30 (4%) | 0.1±0.2 (1%) | 1.9±0.7 (13%) | 1.9±0.7 (9%) |
| Aeoline input | - | - | 0.5±0.5 (8%) | 0.5 (7%) | 0.5±0.5 (5%) | 0.30±0.20 (3%) | 0.2-1.0±0.3 (3-8%) | 0.5±0.5 (3%) | 0.5±0.5 (2%) |
| Submarine ground water | - | - | - | - | 0.6±0.6 (6%) | 0.65±0.54 (6%) | 0.6-3.8±2.0 (8-32%) | 2.3±1.1 (16%) | 0.4±0.3 (2%) |
| Glacial meltwater in polar area | - | - | - | - | - | - | - | 0.3±0.3 (2%) | 0.3±0.3 (1%) |
| Sandy beaches | - | - | - | - | - | - | - | - | 8.3±3.0 (39%) |
| *Total Si input flux* | 7.3 (100%) | 6.2 (100%) | 6.1±2.0 (100%) | 6.7 (100%) | 10.9±4.7 (100%)* | 10.20±1.30 (100%) | 7.8-11.8±2.1 (100%) | 14.8±2.6 (100%) | 21.3±8.5 (100%) |

**Marine Si Sinks**




| | | | | | | | | | |
|---|---|---|---|---|---|---|---|---|---|
| Diatom | 6.0 (100%) | 5.5-7.5 (100%) | 7.1±1.8 (100%) | 6.5-7.4 (100%) | 6.3±3.6 (55%) | - | 7.4-8.8±1.5 (91-100%) | 9.2±1.6 (59%) | 9.2±2.8 (59%) |
| Sponges | - | - | - | - | 3.6±3.7 (32%) | - | 0.04-0.9±0.9 (0-9%) | 1.7±1.6 (11%) | 1.7±1.6 (11%) |
| Reverse weathering | - | - | - | - | 1.5±0.5 (13%) | - | 0 (0%) | 4.7±2.3 (30%) | 4.7±2.3 (30%) |
| *Total Si output flux* | 6.0 (100%) | 5.5-7.5 (100%) | 7.1±1.8 (100%) | 6.5-7.4 (100%) | 11.4±7.6 (100%) | - | 7.4-9.7±1.8 (100%) | 15.6±2.3 (100%) | 15.6±6.7 (100%) |
| Total marine dSi reservoir (Tmol Si) | 97000* | 97000* | 97000 | 97000* | 97000 | 112000 | 112000 | 120000 | 90000 |
| *Marine Si Residence Time (yrs)* | 15000* | 15400* | 15000 | 14000* | 10000 | 12000 | 9500-14400 | 7700 | 4200-7000 |


Based on the evaluations of marine Si sources and sinks, the modern marine Si cycle was long considered to be in a steady state, with an equal input and output flux of Si (see **Table 1** and references therein). However, some recent studies suggest that the dissolution of Si from sandy beaches in the intertidal surf zones contributes a significant flux of dSi to the ocean (Aparicio

et al., 2025; Fabre et al., 2019). This leads to an imbalanced global ocean Si budget (Si input flux: 21.3 Tmol-Si yr$^{-1}$, Si output flux: 15.6 Tmol-Si yr$^{-1}$; see **Table 1**). However, Tréguer et al. (2021) found that Fabre et al. (2019) overvalued the dSi from sandy beaches due to the complex composition of world ocean beaches, variable sand and seawater mixing conditions at both spatial and temporal scales, and the likely overestimation of coastal water dSi concentration (85.4 µmol L$^{-1}$). Furthermore, Si efflux from sandy beaches was included in the submarine groundwater discharge by Cho et al. (2018). Thus, the potential

double counting of the coastal zone's Si input flux may lead to an overestimation of the world ocean's Si sources.

Due to the identification of new pathways and processes, as well as important bSi sinks (e.g., siliceous sponges), the estimation of Si input and output fluxes has doubled since the study by Calvert (1968) (**Table 1**). This has resulted in a shorter residence time of Si in the ocean, from 15 000 years to approximately 7 000 years. Ultimately, more accurate estimates of the biotic and abiotic fluxes and processes that drive the global Si cycle are needed to assess whether it is in a steady state

or if the ocean is becoming more enriched or gradually depleted of dSi. However, this task is challenging, if not impossible,



given the complexity of the large number of interconnected processes, coupled with both natural variability and anthropogenic perturbation.

## 4.3 Interfaces in the oceanic silicon cycle: processes, pathways, and perturbations

Beyond the open ocean, the global Si cycle is shaped by dynamic exchange zones — interfaces where land, sediments, and ice
meet the sea. These interfaces host complex physical, chemical, and biological processes that modulate the pathways and reactivity of Si entering or leaving the ocean. Exploring these often-overlooked boundaries reveals critical controls on Si availability and ecosystem function and is essential to constrain the spatial heterogeneity and long-term evolution of the marine Si cycle.

### 4.3.1 Land–ocean interface: biogeochemical filters and emerging pathways in the silicon cycle

The land-ocean interface plays a critical role in the global Si cycle, serving as a dynamic transition zone where terrestrial inputs, biological transformations, and sedimentary processes interact to modulate Si fluxes to the coastal and open ocean. Rivers, the primary source of Si to the ocean, transport Si derived from continental weathering, with an increasing number of studies using germanium (Ge)/Si ratios, and Si and Ge isotopes to understand Si cycling during weathering and Critical Zone processes on land (Baronas et al., 2018; Fernandez et al., 2022; Frings et al., 2016, 2021). Recent studies have identified a
number of previously overlooked Si sources, including submarine groundwater (Rahman et al., 2019; Santos et al., 2021; Zhu et al., 2025), salt marshes (Williams et al., 2022), and sandy beaches (Aparicio et al., 2025). These new sources challenge the conventional view that land-ocean inputs are dominated by silicifier-modulated riverine fluxes, suggesting that physical processes may play a larger role than previously assumed. The effect of submarine groundwater discharge and beach sand dissolution on the productivity of silicifiers is yet to be investigated.

Silicifying organisms exert substantial control over Si retention, recycling, and export from land. For example, diatoms play a key role in Si cycling in large estuaries, making them effective Si filters that sequester a significant fraction of terrestrial Si before it reaches the ocean (Baronas et al., 2016; Chong et al., 2014; Zhang et al., 2020a). Similarly, plant Si accumulation exerts a major control on the terrestrial Si cycle (Alexandre et al., 1997; de Tombeur et al., 2020), with probable consequences for the land-to-ocean transfer of dSi (Conley, 2002; Conley and Carey, 2015). In fact, the annual soil-plant systems recycle
much more Si (global annual Si uptake by vegetation ranges from 60 to 200 Tmol Si yr$^{-1}$; Conley, 2002) compared to what is transferred from land to oceans through rivers (**Table 1**). Ge and Si isotope mass balances also indicate 12-54% of all Si released during terrestrial weathering is taken up by terrestrial vegetation (Baronas et al., 2018; Frings et al., 2021). Such phenomena are often referred to as the "terrestrial ecosystem silica filter" (Struyf and Conley, 2012). In addition to retaining and enhancing the formation of secondary weathering products on land (Baronas et al., 2020; Cornelis and Delvaux, 2016),
plants also increase silicate weathering rates through a wide range of processes (i.e. bio-weathering; Wild et al., 2022). In contrast to the numerous studies on marine and terrestrial silicifiers, there have been relatively few studies of lacustrine Si





cycle, with most studies focusing on bSi oxygen isotope analyses or palaeoproductivity reconstructions (Frings et al., 2024; Meister et al., 2024). As a result, the impact of freshwater bSi cycling on the land-to-ocean Si flux is relatively unconstrained.

Anthropogenic activities, including land use change, eutrophication, and damming, are reshaping Si fluxes at the land-ocean interface, with implications for diatom productivity and coastal ecosystem health (Laruelle et al., 2009; Maavara et al., 2014). New evidence continues to demonstrate the effects of eutrophication and damming on Si trapping in both river channels and reservoir systems (Ran et al., 2022). Zhang et al. (2020b) showed that damming and eutrophication have altered the Si isotopic signature in the Baltic Sea due to shifts in diatom-driven Si utilization and sedimentary Si regeneration. Additionally, research has shown that coastal Si cycling is amplified by oyster aquaculture, which drives rapid recycling of dSi to the water

column (Ray et al., 2021). Climate change also impacts Si dynamics at the land-ocean interface. Palaeo-records from Chesapeake Bay, the largest estuary in the US, indicate that estuarine Si cycling is sensitive to changes in sedimentation rate over the Holocene, influenced by both climatic shifts (e.g. sea level rise, runoff changes) and human activities (e.g. deforestation) (Nantke et al., 2019). Geilert et al. (2023) demonstrated how climatic events such as coastal El Niño can accelerate marine silicate alteration, driving rapid authigenic clay formation that modifies Si availability in continental margin

sediments. This underscores the sensitivity of coastal Si cycling to episodic disturbances, which are becoming more frequent due to climate change.

### 4.3.2 Benthic interface: silicon recycling and burial at the seawater–sediment boundary

The flux and efficiency of bSi burial in marine sediments primarily depends on 1) the bSi delivery rate to the sediment (itself a function of productivity and pelagic recycling rates; (Van Cappellen et al., 2002); 2) the type of bSi (e.g. diatom, Rhizaria,

sponges, etc.) and its inherent solubility, surface area, and dissolution rates (Aller and Wehrmann, 2025; Maldonado et al., 2019, 2022); 3) detrital sediment supply and overall sediment accumulation (Loucaides et al., 2010; Presti and Michalopoulos, 2008); and 4) diagenetic and redox conditions in marine sediments (McManus et al., 1995; Van Cappellen and Qiu, 1997; Zhu et al., 2024). Regarding the contribution of different organisms to the benthic bSi burial flux, the long-term general consensus of diatom dominance has recently been challenged by a number of studies on other silicifying organisms, notably rhizarians

and sponges. Maldonado et al. (2019) have shown that, due to the high preservation efficiency (45.2 ± 27.4%), sponge spicules could contribute a burial flux of 1.71 ± 1.61 Tmol Si yr$^{-1}$ globally, more than an order of magnitude higher than previously estimated. While dissolution rates can vary significantly between different sponge types, bSi of the dominant demosponges is orders of magnitude more resistant to dissolution compared to diatom bSi under seafloor-like conditions, although the cause is yet to be identified (Maldonado et al., 2022). While rhizarians have a much lower preservation efficiency of 6.8 ± 10.1%

(Maldonado et al., 2019), a recent study has significantly revised their global productivity upwards to 2-58 Tmol Si yr$^{-1}$ (Llopis Monferrer et al., 2020). Assuming a similar benthic export efficiency to diatoms (33%), rhizarian bSi could thus theoretically contribute 0.1-3.4 Tmol Si yr$^{-1}$ to the benthic bSi burial flux. The lower end of this range is consistent with the 0.09 ± 0.05





Tmol Si yr$^{-1}$ estimated by Maldonado et al. (2019), using direct observations of rhizarian skeletons in marine sediments from across the global ocean.

Once deposited, bSi undergoes (partial) dissolution, often coupled with diagenetic alteration (see the recent review by (Aller and Wehrmann, 2025 and **Fig. 3**). The latter includes the direct alteration of bSi and/or the formation of authigenic silicate (clay) phases, frequently as coatings and partial replacement of the initial bSi phases. Recent studies have attempted to quantify the relative contribution of these different processes by investigating the Si and Ge isotopic signature of both sediments and pore waters (Baronas et al., 2019; Cassarino et al., 2020; Closset et al., 2022). Indeed, unique isotopic

fractionation during these processes could lead to predictable shifts in the Si isotopic composition ($\delta^{30}$Si) of pore water values, helping decipher the complex Si cycle in surface sediments (see **Box 4**). Most of the studies targeted the Southern Ocean or upwelling areas that are known to play a dominant role in the global marine Si cycle, hosting abundant diatom populations and exporting large quantities of bSi. For example, the Southern Ocean produces ~30% of global silica and acting as the largest sedimentary sink (~2 Tmol Si yr$^{-1}$; Tréguer, 2014). The main output of these studies is that most settling bSi dissolves within

the upper few centimetres of the sediment directly below the sediment-water interface, regulating pore water dSi concentrations and deep ocean Si fluxes. For example, in the Southern Ocean, diffusive Si fluxes from the sediment show a zonal trend, peaking near the Polar Front and correlating with sediment bSi content likely reflecting diatom productivity in the overlying water column (Closset et al., 2022). However, isotopic compositions of both pore water and bSi preserved in the sediment indicate that secondary processes such as the formation of authigenic alumino-silicates from the dissolving bSi, dissolution of

reactive lithogenic silica, and adsorption onto particles can play a significant role but differ between regions (Frings et al., 2024).

When authigenic silicate clay formation incorporates major cations, it lowers the pH of surrounding pore waters and therefore reduces seawater alkalinity, which is termed "reverse weathering" (Mackenzie and Garrels, 1966). It is important to recognize that not all authigenic mineral formation results in reverse weathering, but only that which consumes alkalinity and

therefore drives an increase in atmosphere-ocean pCO$_2$ (Hong et al., 2025). While there is ample evidence that clay authigenesis is a major sink of Si and some trace elements (Baronas et al., 2019; Läuchli et al., 2025; Rahman et al., 2017), only a few studies have directly evidenced the associated consumption of cations (Michalopoulos and Aller, 1995, 2004; Wu et al., 2025). Nevertheless, it has recently been proposed that reverse weathering has played a crucial role in stabilizing marine pH and regulating climate over geological timescales (Isson and Planavsky, 2018). Despite its importance, the formation of

authigenic (newly precipitated) clays is difficult to detect in sediments dominated by detrital clays but it can occur relatively quickly (less than one year; Michalopoulos et al., 2000). However, a re-estimate of both forward and reverse weathering rates in marine sediments has suggested that there is no net alkalinity flux in marine sediments, and both processes must be tightly coupled to enable each other (Wallmann et al., 2023). Furthermore, forward and reverse silicate weathering exhibits only a secondary role, and is strongly influenced by organic matter degradation and carbonate mineral reactions in marine sediments

(Hong et al., 2025). Therefore, the role that bSi diagenesis-enabled reverse weathering plays in controlling Earth's long-term carbon cycle balance is still open for debate.




### 4.3.3 Cryospheric interface: the role of glacial and sea-ice systems in silicon cycling

The cryosphere includes all of Earth's frozen systems, both on land and in the ocean. This encompasses snow, glaciers, ice sheets, icebergs, sea ice, frozen rivers and lakes, permafrost, and seasonal frozen grounds. While the role of each of these
systems on the Si cycle is not yet fully understood, significant progress has been made in studying glaciers, ice sheets, sea ice and permafrost (Alfredsson et al., 2015; Hatton et al., 2019b; Hawkings et al., 2017; Pokrovsky et al., 2013).

Glaciers and ice sheets cover up to 30% of Earth's land during glacial cycles, releasing significant amounts of freshwater and sediment impacting downstream ecosystems (Hopwood et al., 2025; Tréguer, 2014). This discharge can limit light penetration into the water column (Halbach et al., 2019) but it also supplies essential nutrients to marine environments,
increasing primary production (Hawkings et al., 2017). Recent studies highlighted the significant contribution of glaciers and ice sheets to the global Si cycle, supplying both dSi and amorphous silica (aSi) to marine environments (Hawkings et al., 2017; Meire et al., 2016; Tréguer et al., 2021). These contributions rival other inputs such as dust deposition, groundwater discharge and hydrothermal input (Hawkings et al., 2017). The magnitude and reactivity of glacial Si fluxes have a direct influence on the dissolution rates and availability of bioavailable dSi, potentially enhancing primary production in coastal and shelf seas
(Hatton et al., 2023). Pro-glacial rivers are characterized by a lighter isotopic signature in both dSi and aSi, lower than the average of non-glacial rivers, from 0.16‰ to 1.38‰ respectively for the dissolved fraction (Hatton et al., 2019b) and lower than the surrounding sediment and bedrock (Hawkings et al., 2018). Under glacial conditions, enhanced chemical weathering — driven by processes such as dissolution–reprecipitation and the leaching of freshly abraded mineral surfaces — produces highly reactive inorganic aSi. Long water residence times and intense erosion further promote silicate dissolution, even at low
temperatures (Hatton et al., 2021). Additionally, suspended fine particulate matter from intense erosion carries a substantial fraction of dissolvable aSi (Baronas et al., 2021; Hatton et al., 2019a; Hawkings et al., 2017). This highly soluble aSi can represent up to 95% of total potentially bioavailable silica dSi and aSi in areas receiving glacial meltwaters. Up to 20% of this aSi can dissolve rapidly after two weeks in low-nutrient seawater (dSi < 1μM) at 18°C (Hawkings et al., 2017; Hendry et al., 2025). An even less studied and potentially highly bioavailable, colloidal-nanoparticulate size Si (CNSi) seems to represent a
significant fraction of total bioavailable Si, only present in pro-gracial rivers, which may be a similar order of magnitude to the concentration of dSi and aSi (Pryer et al., 2020). Dissolution, precipitation, sedimentation, resuspension and uptake are key processes in glacial estuary environments controlling the fate of glacial dSi and aSi inputs to the coastal environment (Hendry et al., 2025). In addition, complex, heterogeneous and still poorly understood processes in pro-glacial environments and fjords lead to lower dSi concentrations in interstitial fluids near glacial outputs than in coastal shelf sediments, which may
impact the dSi cycle in the water column and its offshore dynamics (Hendry et al., 2025; Ng et al., 2020, 2022).

Sea ice consists of frozen seawater floating on the ocean's surface, and it can be formed seasonally or be perennial. During its formation, small pockets of brine are formed due to salt expulsion (Eicken, 2003). As the expulsion of brine increases the salinity of the mixing layer, it can enhance sea water circulation (Wang and Danilov, 2022), therefore promoting the upwelling of deep water and affecting the nutrient supply to surface waters. Furthermore, sea ice supports diverse communities, including





diatoms, inhabiting these brine pockets, as well as the sea ice surface and attached underneath the sea ice (Arrigo, 2017). For example, by focusing on the distribution of natural Si isotopes within the Antarctic pack ice, Fripiat et al. (2014) found that bSi dissolution contributes significantly to silica production, with a dissolution-to-production ratio between 0.4 and 0.9. This suggests that there is considerable regeneration of Si within the brine network, which fuels new bSi production. This study was crucial in filling a gap in the understanding of sea ice Si cycling, a topic that is still poorly studied due to the difficulty of

measuring rates in such environments.

The role of permafrost in the polar regions and their influence on the Si cycle remains uncertain, particularly in the context of a warming climate. Seasonal isotopic variations observed in the Lena River (Russian Siberia) suggest that Si sources shift throughout the year—clay minerals dominate in winter, while summer brings increased dissolution of silicate minerals and phytoliths from the upper active layer (Sun et al., 2018). These findings imply that permafrost thawing, driven by climate

change, could significantly alter Si fluxes and isotopic compositions in high-latitude rivers. However, a study by Mavromatis et al. (2016) in the Yenisey River and its tributaries found that mineral dissolution in the watershed plays a crucial role in defining the riverine dissolved $\delta^{30}Si$, regardless of permafrost coverage. Their findings highlight the influence of suspended particles (e.g. clays), plant material decomposition (both contributing lower $\delta^{30}Si$), and groundwater discharge enriched in heavier isotopes due to secondary silicate formation. These contrasting findings underscore the complexity of Si cycling in

regions covered by permafrost where multiple processes occur. Further studies are essential for predicting how ongoing climate change might reshape the Si export to the ocean and how these signals can be interpreted in palaeorecords.

In a recent study, Opfergelt et al. (2024) investigated how winter river ice formation in the Lena River affects the biogeochemical cycling of key nutrients like nitrogen and Si, and their transport to the Arctic Ocean. During very cold periods, *frazil ice* forms microzones that slow water flow and create favourable conditions for microbial activity. These zones allow

organic matter recycling and alter nutrient processing. The formation of ice crystals leads to the precipitation of aSi, which changes $\delta^{30}Si$ and the Ge/Si ratio in the remaining water. Through isotope mass balance, Opfergelt et al. (2024) estimate that about 39% of Lena River water passes through this ice-induced filter during winter, leading to notable silica loss and a shift in river biogeochemistry.

Climate change is accelerating the melting of glaciers, permafrost and sea ice, reducing ice-covered areas and increasing

open-water regions. These shifts alter habitats, species distributions, and nutrient dynamics. The melting of sea ice, glaciers and ice sheets also releases previously trapped trace metals and nutrients into seawater, further influencing marine productivity and biogeochemical cycles (Hopwood et al., 2020). Given their vast size and prolonged melting periods, ice sheets may have a more significant global impact on Si input compared to smaller glaciers with more seasonal contributions. While glacial retreat and melting are expected to increase Si fluxes to the ocean, the long-term effects remain uncertain, as they could reshape

marine ecosystems and biogeochemical cycles (Hatton et al., 2019b; Pryer et al., 2020). Palaeorecords of $\delta^{30}Si$ and Ge/Si signatures in sedimentary bSi are emerging as a useful tool to reconstruct and quantify the effects of glaciation on the Si cycle and weathering (Hou et al., 2025). Understanding the role of glaciers in the Si cycle — particularly how their contributions interact with other sources and evolve under future climate scenarios — remains a key area of research.





## 5. Perspectives and conclusions

Despite major advances in our understanding of silicification, silicifier diversity, and the global Si cycle, critical knowledge gaps persist across taxonomic, physiological, ecological, and geochemical dimensions. To move the field forward, we outline key research priorities that we believe will be pivotal in developing a more integrated and predictive understanding of biosilicification and its role in the marine Si cycle. Addressing these challenges will require integrative and interdisciplinary research approaches that bridge molecular biology, ecology, biogeochemistry, (palae)oceanography, and Earth system science.

**1. Expand taxonomic and functional knowledge across silicifiers:** Research remains heavily biased toward a few model organisms such as diatoms and terrestrial plants. Many other silicifying taxa, including Rhizaria, silicoflagellates, sponges, macrophytes, and even some prokaryotes, are understudied despite their potential significant contributions to the Si cycle. Broader taxonomic coverage, including the development of new model systems and comparative studies, will be essential to capture the evolutionary, functional, and ecological diversity of silicification strategies across domains of life.

**2. Elucidate cellular mechanisms and physiological regulation:** Much remains to be understood about how organisms control Si uptake, intra- and intercellular transport, polymerization, and efflux. For instance, the mechanisms of Si efflux, crucial for homeostasis in diatoms, are poorly characterized in most groups. Investigating the molecular basis of silicification — including gene expression, transporter function, and nanoscale silica assembly — will benefit from the use of high-resolution imaging, omics technologies, and experimental manipulation under variable environmental conditions.

**3. Integrate silicifiers into ecosystem and global models:** Current biogeochemical models often overlook key silicifiers such as sponges and Rhizaria. Their omission limits our ability to predict Si fluxes and understand how Si cycling interacts with other elemental cycles like carbon and nitrogen. Incorporating a broader range of silicifiers into these models is essential, particularly under changing climate regimes.

**4. Clarify interactions between biotic and abiotic Si processes:** The interplay between biological silicification, biogenic
silica dissolution, silicate weathering, and authigenic clay formation remains insufficiently understood. This is especially true in transitional environments like estuaries and coastal margins, where complex feedbacks occur. Further research is needed to disentangle the contributions and temporal dynamics of biotic versus abiotic Si pathways.

**5. Assess environmental and climatic influences on silicification and Si fluxes:** While the environmental drivers of silicification are well studied in diatoms, their effects on other silicifiers are largely unknown. Additionally, how climate-
related stressors — such as warming, acidification, altered precipitation, and sea-level rise — affect Si cycling and silicifiers distribution is not well constrained. These processes may reshape the ecological roles of silicifiers and the fate of Si at critical interfaces such as the land–ocean boundary.

**6. Foster cross-disciplinary integration:** Bridging disciplinary divides — between marine and terrestrial science, biology and geochemistry — will enhance our understanding of the Si cycle. Cross-comparisons (e.g. between plant and algal



silicification), training in diverse analytical techniques, and collaboration across fields will enable more holistic interpretations of both modern processes and palaeoenvironmental records.

Silicon plays a pivotal yet still underappreciated role in shaping biological and geochemical processes across marine and terrestrial systems. This review highlights the remarkable diversity of silicifying organisms and their crucial, but often

overlooked, roles in shaping the global marine Si cycle. From unicellular diatoms to multicellular sponges, and from well-studied phytoplankton to enigmatic protists and prokaryotes, silicifiers contribute to a wide array of ecological functions and influence biogeochemical feedbacks across spatial and temporal scales. As research expands beyond classical models, a more nuanced and integrative picture of the Si cycle is emerging — one that includes diverse silicifiers, complex environmental feedbacks, and deep-time dynamics. Despite significant progress over the last decade, major uncertainties persist regarding

the cellular machinery of silicification, the environmental controls of Si fluxes and the integration of silicifiers into predictive models. These gaps are especially pressing in the context of global change, where shifts in climate and nutrient regimes are likely to affect the distribution, abundance and ecological function of silicifiers in ways that are still poorly understood. Addressing these challenges will require coordinated, cross-disciplinary efforts that link biological, ecological and geochemical research. Leveraging recent advances in molecular biology, high-resolution imaging, in situ monitoring, and

Earth system modeling will be key to unraveling the complexity of Si cycling. Advancing our understanding of biosilicification is not only essential for reconstructing past marine environments but also for predicting the future of ocean biogeochemistry in an era of rapid planetary change.

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

**Financial support.** Félix de Tombeur received funding from the European Union's Horizon 2020 research and innovation programme under the Marie Skłodowska-Curie grant agreement No. 101021641 (project SiliConomic). Alessandra Petrucciani received funding from the European Union's Horizon research and innovation actions programme under grant agreement No. 101083355 (project DESIRED). Natasha Bryan received funding from the Helmholtz Association through INSPIRES Project 'Polar-Flow'. María López-Acosta received funding by a postdoctoral fellowship and project funded by the 'Xunta de Galicia' 2005 (IN606C-2023/001). Antonia U. Thielecke received funding from the Helmholtz Association through the Helmholtz Young Investigator group 'Side-Effect' (VH-NG-1600). Natalia Llopis Monferrer received funding from the European Union's Horizon Europe research and innovation program under grant agreement No. 101064167 (MSCA postdoctoral fellowship Si-ORHIGENS). Dongdong Zhu was supported by the China Postdoctoral Science Foundation under Grant No. 2024M753048, and the Qingdao Postdoctoral Grant (QDBSH20240202136).






## Boxes

**Box 1**: **Studying biosilicification and silicon uptake: from molecular tools to radioisotope enrichment physiological experiments**

Biosilicification is the molecular process by which organisms take up silicic acid and transform it into resistant, elastic biomaterials. The study of this process is crucial not only for understanding its biological and ecological significance, but also for unlocking sustainable solutions in material sciences. At the same time, insights gained at the molecular and cellular levels enhance our understanding of the marine silicon cycle and improve the accuracy of global biogeochemical models. Recent advances in molecular and single-cell techniques are pushing the frontiers of this field. This toolbox highlights the key technologies currently used to investigate biosilicification.

A central focus in biosilicification research is the identification of molecular markers that signal the presence of this process. Among these, silicon transporters are particularly informative. Due to their highly conserved structure (Hildebrand et al., 1997), these proteins are frequently identified using sequence similarity searches, hidden Markov models, and phylogenetic analyses to detect them in genome and transcriptome datasets (e.g. Durkin et al., 2016; Marron et al., 2016). Functional annotation and expression analyses can help confirm their roles in silicon transport and their regulation across different environmental conditions and taxa. However, silicon transporters are not necessarily the only proteins involved in this complex process. An indirect approach to studying silicification-related genes is to compare RNA transcript levels under limiting and excess silicate availability. By correlating the transcription of unknown genes with that of known silicifying proteins (e.g. Maniscalco et al., 2022; Thamatrakoln and Hildebrand, 2008), it is possible to identify potential candidates involved in silica metabolism. However, this approach is limited to the organisms that can be cultured or maintained in controlled conditions. Beyond genetic analyses, tools such as AlphaFold (Abramson et al., 2024; Mirdita et al., 2022) provide accurate structural predictions of proteins (Jumper et al., 2021). By modelling protein structures, it is possible to study active sites, binding interactions with silica precursors, and the molecular mechanisms driving silica deposition (Knight et al., 2023). Computational simulations and molecular docking further refine these insights by examining protein interactions with silicic acid and other biomolecules. In addition, structural alignment algorithms such as the Foldseek cluster (Barrio-Hernandez et al., 2023) enable large-scale comparisons, helping to identify domain families and detect remote structural similarities, shedding light on protein function and evolution in diverse organisms. Integrating these predictive approaches with experimental techniques such as electron cryomicroscopy, X-ray crystallography, and biochemical assays strengthens structural interpretations, deepening our understanding of biomineralization processes.

There are also qualitative methods for studying silicification, such as the use of fluorescent compounds. More recent developments have used PDMPO ((2-(4-pyridyl)-5-[(4-(2-dimethylamino-ethylaminocarbamoyl)methoxy)-phenyl]oxazole) in sponges, diatoms, and siliceous Rhizaria (McNair et al., 2015; Ogane et al., 2009; Schröder et al., 2004).





This compound is incorporated into newly polymerized biogenic silica under acidic conditions and emits a strong green fluorescence allowing for single-cell and mortality-independent information on silicification and growth rates (Leblanc and Hutchins, 2005; McNair et al., 2015; Shimizu et al., 2001). While this technique is not fully quantitative, these fluorescent markers provide valuable insight into the silicification process, helping to determine the spatial and temporal patterns of silica deposition and offering a deeper understanding of biomineralization dynamics.

The study of radioisotopes and stable isotopes also provides valuable insights into the dynamics of silicification, including uptake mechanisms, intracellular processing, and environmental influences. Silicon stable isotope tracing ($^{30}$Si) has long been used to simultaneously quantify biogenic silica production and dissolution in the ocean (Fripiat et al., 2007; Nelson and Goering, 1977). Despite the long sample preparation times and complex instrumental analysis required, they offer a powerful tool to understand and quantify nutrient uptake mechanisms, especially when used in conjunction with other stable isotopes such as carbon ($^{13}$C) or nitrogen ($^{15}$N-nitrate or $^{15}$N-ammonium). Using secondary ion mass spectrometry, it is possible to obtain direct measurements of single cell uptake rates from the same sample (Olofsson et al., 2019). Due to its easy applicability and increased sensitivity compared to stable isotopes, silicon radioisotope tracing ($^{32}$Si) has gained increasing popularity in recent decades (Closset et al., 2021; Giesbrecht and Varela, 2021; Krause et al., 2019). It has been successfully applied to a variety of silicifying organisms, beginning with diatoms and later extended to others like Rhizaria (Llopis Monferrer et al., 2020). This technique may also be applicable to other organisms such as silicoflagellates and sponges, that can be maintained under controlled conditions.

**Box 2**: **Siliceous microfossils as markers of past and present oceanic conditions**

Siliceous microfossils are widely used to reconstruct past environments, from deep geological times to the recent past. Among these, radiolarians, diatoms, and dinoflagellate cysts are particularly valuable for biostratigraphic studies, especially in environments where carbonate fossils are scarce or poorly preserved (Barron et al., 2014; Carvajal-Landinez et al., 2024; Iwai et al., 2025). Siliceous microfossils also serve as proxies in palaeoclimatic reconstructions, providing insights into water masses and environmental parameters such as sea ice cover, sea surface temperature (SST), and marine productivity (Torricella et al., 2025).

Diatoms play a key role in palaeoceanographic and palaeoclimatic reconstructions, particularly at high latitudes, where carbonate preservation is poor. Their distribution is controlled by surface water parameters including light availability, salinity, nutrient levels, temperature, sea ice extent, and the concentration of dissolved silicic acid (Torricella et al., 2022). Benthic diatoms are also influenced by substrate type and water depth, typically flourishing at around 100 meters where sunlight still reaches the seafloor (Round et al., 1990). Because diatom species occupy well-defined ecological niches, their fossil assemblages can be used to infer past oceanographic variability. However, only about 10% of surface-dwelling



diatoms are preserved in sediments. Their preservation is affected by silica dissolution during settling and at the seafloor, as well as by lateral transport, bioturbation, grazing, and diagenesis (e.g. Crosta and Koç, 2007; Ran et al., 2024). Despite these challenges, diatom assemblages provide crucial information on past surface ocean and climate conditions across a range of regions, including the Southern Ocean (e.g. Crosta et al., 2021), the Arctic (e.g. Oksman et al., 2019), and mid-latitudes (e.g. Bárcena et al., 2001). Exceptionally well-preserved, laminated diatom oozes — particularly those found in Southern Ocean coastal sites — enable high-resolution reconstructions of palaeoceanographic conditions (e.g. Alley et al., 2018; Leventer et al., 2006; Roseby et al., 2022). Seasonally laminated, diatom-rich sequences allow for annual-scale investigations of productivity and sedimentation processes (e.g. Leventer et al., 1993; Maddison et al., 2006; Pike and Kemp, 1997). In contrast, laminations formed by giant diatoms are often associated with specific oceanographic settings such as frontal systems and nutrient trapping, rather than seasonal cycles (Kemp et al., 2006). Stable isotope analyses have recently been applied to diatoms to strengthen palaeoenvironmental interpretations. Commonly measured isotopes include carbon and nitrogen in diatom-bound organic matter, as well as oxygen and silicon in the siliceous frustules (Frings et al., 2024; Robinson et al., 2020; Swann et al., 2010). The oxygen isotope $\delta^{18}O$ provides information on ice volume, temperature, and regional oceanographic conditions (e.g. Hodell et al., 2001; Shemesh et al., 2001), while $\delta^{30}Si$ reflects the degree of dissolved silica utilization by comparing uptake rates to ambient concentrations (Cardinal et al., 2005; De La Rocha et al., 1998; Varela et al., 2004). In addition, $\delta^{15}N$ and $\delta^{13}C$ serve as proxies for past ocean productivity (Crosta and Shemesh, 2002; Schneider-Mor et al., 2005).

Radiolarians, like diatoms, are widely employed in modern and palaeoceanographic studies due to their sensitivity to changes in SST, thermocline depth, nutrient availability, and water mass distribution (Hernández-Almeida et al., 2017; Zhang et al., 2015, 2018). Their siliceous skeletons are well preserved in sediments and form distinct assemblages that record past temperature gradients, upwelling intensity, and ocean stratification (e.g. Civel-Mazens et al., 2024). Radiolarians can also serve as valuable palaeoceanographic proxies in regions where other silicifiers, such as diatoms, are rare or poorly preserved, particularly in certain deep-sea or oligotrophic sedimentary environments. Several studies have used both diatom and radiolarian assemblages to quantitatively estimate past sea temperature, salinity, and sea ice extent using transfer function techniques (Chadwick et al., 2022; Civel-Mazens et al., 2023; Crosta and Koç, 2007). These functions are based on three key datasets: (1) the modern species distributions in core-top sediments or water samples, (2) contemporary oceanographic parameters obtained from in situ measurements, and (3) fossil assemblages from sediment cores. A major limitation of this approach is the requirement for comprehensive coverage of all three datasets within the target study region.

Isotopic analyses can also be applied to radiolarians. The silicon isotope $\delta^{30}Si$ is used to reconstruct past dissolved silica concentrations (Doering et al., 2021), while $\delta^{18}O$ serves as a tracer for water temperature. Robinson et al. (2015) tested $\delta^{15}N$ measurements on radiolarian tests, finding that it reflects the isotopic composition of their food sources. However, $\delta^{15}N$ in radiolarians shows significant offsets compared to other siliceous microfossils, emphasizing the need to isolate radiolarian



fractions carefully from other organisms. Abelmann et al. (2015) observed that $\delta^{18}O$ and $\delta^{30}Si$ from radiolarians and diatoms may not always align with changes in assemblages, but still offer valuable insights into silica uptake by different groups and changes in surface and subsurface water masses (down to ~400 m).

Silicoflagellates, though less abundant and less studied than diatoms and radiolarians, are still useful in micropalaeontology as proxies for palaeotemperature and palaeoceanographic reconstructions (e.g. McCartney et al., 2022; Rigual-Hernández et al., 2016; Torricella et al., 2021). Sponges also contribute to siliceous microfossil records, with their spicules commonly preserved in marine sediments. Although assigning isolated spicules to specific sponge taxa is often difficult and sponge-derived material may be scarce in some environments such abyssal plains, they have been used to infer evolutionary, ecological, and environmental conditions (e.g. De Freitas Oliveira et al., 2020; Sim-Smith et al., 2017). Koltun (1960) first emphasized their utility in reconstructing salinity, temperature, and depth from sedimentary sequences. Sponge spicules now serve as valuable proxies for reconstructing a range of palaeoenvironmental conditions, including water flow regimes and velocities (Kuerten et al., 2013), pH (Pisera and Sáez, 2003), light availability (Harrison, 1974), temperature (Gaino et al., 2012), currents (Molina-Cruz, 1991), salinity (Cumming et al., 1993), and water depth (Łukowiak, 2016). Stable isotope analyses, particularly those focusing on $\delta^{30}Si$, are increasingly used on sponge silica to investigate silicon cycling in ancient oceans (Egan et al., 2012). Sutton et al. (2011) demonstrated that $\delta^{30}Si$ in sponge spicules offers a more accurate and integrated view of whole-ocean silicon cycling than surface-dominated diatom records alone.

In conclusion, siliceous microfossils offer a diverse and powerful set of tools for palaeoenvironmental and palaeoclimatic reconstruction in a range of marine environments. Each group contributes unique ecological and geochemical information: diatoms, silicoflagellates, and radiolarians are particularly valuable for deciphering past surface and subsurface oceanic conditions, while sponge spicules provide complementary information on deeper water mass properties, benthic environments, and the Si cycle. Recent advances in stable isotope geochemistry have further expanded the potential of these microfossils, allowing for increasingly precise reconstructions of nutrient dynamics, temperature variations and Si utilisation.

**Box 3: Trait-based approaches to understand the functions of silicification: are diatoms and plants the same?**

Why do species or families invest more in silicification than others? Does silicification involve trade-offs with key ecological strategies? What are the costs and benefits of silicification? In land plants, integrating Si concentrations in trait-based approaches has recently proved useful to answer those fundamental questions about Si utilization (de Tombeur et al., 2023b). The question arises as to whether the same approaches could be used in marine silicifiers such as diatoms. In plants, measuring organ-scale Si concentration is relatively easy and offers a reasonable proxy for the degree of silicification, which can then be linked to other functional traits (de Tombeur et al., 2023a, 2025). In diatoms, we hypothesize that it is probably



more the shape of silica-based skeletons that is linked to specific functions and associated trade-offs, and less the bulk diatom Si concentrations per se. As such, using only diatom Si concentrations, as done in plants, may not be sufficient to infer specific functions associated with silicification given the large diversity of silicification patterns. Instead, diatom shapes should be involved in current trait-based frameworks. Beyond silicification, current trait-based approaches in diatoms mostly rely on metabolic traits (e.g. nutrient uptake rates; Ács et al., 2019; Litchman, 2022), and less on morphological and chemical traits that are reasonably easy to measure, as it the case in plants (e.g. leaf thickness, leaf area, leaf nitrogen concentration). This major difference explains why trait-based approaches to better understand the role and evolution of silicification will probably not follow the same paths for diatoms and plants. Maintaining a constant dialog between trait-based ecologists working on both types of silicifiers is, however, essential.

**Box 4**: **Silicon natural, stable isotopes as tracers of the marine silica cycle**

In the natural environment, silicon (Si) has three stable isotopes: $^{28}Si$, $^{29}Si$, and $^{30}Si$. The natural isotopic composition, referred to as $\delta^{30}Si$ when expressed as the $^{30}Si/^{28}Si$ ratio relative to a reference standard, serves as a powerful tracer for elucidating the marine Si cycle by recording key processes such as biological uptake, dissolved silica (dSi) utilization, and diagenetic alteration. In the surface ocean, diatoms preferentially incorporate $^{28}Si$ when building their opaline frustules, leaving the residual dSi enriched in $^{30}Si$ (De La Rocha et al., 1997; Sutton et al., 2013). This isotopic fractionation (usually noted $^{30}\varepsilon$ or $\Delta^{30}Si$) offers a window into dSi utilization: higher $\delta^{30}Si$ values in diatoms signal regions where Si is strongly depleted and efficiently used (Grasse et al., 2020). Consequently, measuring $\delta^{30}Si$ in sedimentary diatom frustules allows to reconstruct past shifts in nutrient availability and productivity, thereby shedding light on glacial-interglacial transitions and long-term variations in the Si cycle (e.g. De La Rocha et al., 1998; Doering et al., 2016; Egan et al., 2012).

In addition to diatoms, other silicifiers, such as sponges and radiolarians, provide complementary insights into Si cycling across various marine environments. Sponges, as benthic organisms, record Si isotope signatures from deeper water masses. Studies have demonstrated that isotopic fractionation in sponge spicules correlates closely with ambient dSi concentrations, and this relationship appears to be independent of factors like temperature, pH, salinity, or other nutrient levels (Hendry et al., 2010; Hendry and Robinson, 2012; Wille et al., 2010). Radiolarians, on the other hand, are siliceous protozooplankton that inhabit both surface and deep waters unveiling the potential to reconstruct past dSi concentrations from different water depths (Fontorbe et al., 2016, 2017; Hendry et al., 2014). Current estimates of the isotopic fractionation of radiolarians rely on comparing $\delta^{30}Si$ values from radiolarians in surface sediments with those of the surrounding water mass (Abelmann et al., 2015; Doering et al., 2021).

Beyond these biological processes, $\delta^{30}Si$ is also a powerful tool for tracing terrestrial inputs and sedimentary recycling. Generally, silicate mineral weathering on land releases dSi with elevated $\delta^{30}Si$ compared to primary silicate minerals, due



to the formation of isotopically light phases, such as secondary minerals in soils and biogenic phytoliths in vegetation (Baronas et al., 2018; Frings et al., 2021; Ziegler et al., 2005). Within marine sediments, the combined effects of biogenic silica dissolution, silicate mineral dissolution, and authigenic clay formation are also reflected in the isotopic composition of porewaters and the buried silica pool (Closset et al., 2022; Ehlert et al., 2016; Geilert et al., 2020, 2023; Ng et al., 2022). Taken together, these processes underscore the versatility of $\delta^{30}Si$ as a geochemical tracer for both modern and ancient oceanic Si dynamics.

**Fig. 4: Schematics of the silicon (Si) cycle and isotope fractionation during various oceanic processes, using diatoms as a model for the biogenic silica (bSi) component. The dissolved silica (dSi) pool is depicted by red dots, with its isotopic composition represented by yellow squares. The bSi and detrital silicate pools are illustrated with representative symbols (diatoms, sediments, rocks), and their isotopic composition are shown as orange squares.**





Various processes in the ocean affect isotopic fractionation throughout the Si cycle (**Fig. 4**). During the production of bSi, such as in the case of diatoms in surface waters, organisms preferentially incorporate lighter Si isotopes, raising the isotopic composition (high $\delta^{30}$Si value) of the surrounding seawater. Simultaneously, the bSi itself becomes enriched in lighter Si isotopes (low $\delta^{30}$Si value; **Fig. 4 - 1**). At the end of the productive season, the bSi, characterized by its lower $\delta^{30}$Si value compared to dSi, is exported to deeper oceanic layers (**Fig. 4 – 2**). In the deep ocean, bSi undergoes gradual dissolution, releasing its low $\delta^{30}$Si signature back into the surrounding seawater. It remains uncertain whether this dissolution process actively fractionates Si isotopes — specifically, whether the release preferentially favours lighter Si isotopes (as proposed by Demarest et al., 2009) or not (as suggested by Wetzel et al., 2014). Nevertheless, the overall effect of this recycling process is the maintenance of a low $\delta^{30}$Si signature in deep waters (**Fig. 4 – 3**). The light $\delta^{30}$Si imprint from the deep waters is subsequently transported back to the surface through vertical mixing processes, particularly during winter periods or in upwelling regions, where it becomes available for uptake by new primary producers (**Fig. 4 – 4**). Simultaneously, the bSi that does not undergo further recycling within the water column settles and reaches the sediment (**Fig. 4 – 5**). Upon deposition in the sediment, bSi remains stored over extended timescales, during which it continues to undergo dissolution, releasing its light isotopic composition into the pore or interstitial waters (**Fig. 4 – 6**). These Si-rich pore waters with low $\delta^{30}$Si value eventually diffuse back into the overlying bottom waters, contributing to the Si pool in the deeper ocean layers (**Fig. 4 – 7**). Under certain conditions, the dSi in pore waters may precipitate back into a mineral phase, forming authigenic clays. While the mechanisms underlying this process remain poorly understood, evidence suggests that it may involve isotopic fractionation, preferentially incorporating the lighter Si isotope into the newly formed clay minerals, thereby increasing the $\delta^{30}$Si of the dissolved phase (**Fig. 4 – 8**).
