# Peer review of "Silicification in the Ocean: from molecular pathways to silicifiers'"

_EGUsphere, 2025_

## Author Comment (AC1)

**Response to Reviewer #1**

Many thanks for the opportunity to review "Silicification in the ocean: from molecular pathways to silicifiers' ecology and biogeochemical cycles" by Closset et al. This paper was conceived and written by a collaborative of early career researchers, and I have to congratulate them on the initiative and well-written paper. The manuscript represents a useful summary of the state-of-the-art on the subject of marine silicification, which is highly relevant for Ocean Sciences, finishing with some key priorities for future research. I enjoyed reading it, and I think it will be a useful resource for the community. The scientific significant and quality are excellent, and presentation quality is good. As such, and subject to just a few moderate revisions, I think that this manuscript will be highly suitable for publication in the Ocean Science Jubilee: reviews and perspectives special issue.

**AUTHORS' RESPONSE:**

We would like to sincerely thank Reviewer 1 for their thoughtful and supportive evaluation of our manuscript "Silicification in the ocean: from molecular pathways to silicifiers' ecology and biogeochemical cycles". We are very grateful for your positive comments on both the quality and relevance of our work, as well as for recognizing the collaborative effort of early career researchers behind this contribution.

We are pleased that you found the manuscript well-written, scientifically significant, and a valuable resource for the community. Your encouraging feedback is highly motivating for us as we continue to establish ourselves in the field.

In the following, we provide detailed responses to the moderate revisions you suggested and explain how we have addressed each point in the revised version of the manuscript.

**My main comments are to:**

'smooth' over the paper, to make sure ideas and concepts are introduced at appropriate points; include further discussion on the use of stable silicon isotopes for the study of silicification processes.

**Manuscript structure**

I found some of the manuscript slightly out-of-order with respect to the overall structure of the paper. In particular there are some terms defined in Section 3, and some ideas introduced regarding diatoms, that have already been referenced (e.g., "frustule" is defined on line 397, but is used in an earlier section of the paper). I understand that different sections of the manuscript will have been written by different authors (especially in a large review as is the case here) but I would suggest that a subset of the authors have a review of the manuscript to make sure that everything appears in a sensible order. (It might well be that some of the information in Section 3 is somewhat superfluous and could simply be trimmed out).

**AUTHORS' RESPONSE:**

We thank the reviewer for their input. We have gone through the manuscript and made minor adjustments throughout to ensure abbreviations are used consistently, definitions are introduced and to remove redundancy where necessary at appropriate points. For example:

Line 139: Removed the following as it is better introduced in line 224 "Sponges take up dSi from seawater through their outer epithelial cells and transport it internally across multiple cell layers to specialized silica-secreting cells called sclerocytes. Within sclerocytes, dSi is concentrated in silica deposition vesicles (SDV), where polymerization occurs to form silica spicules, the structural elements of the sponge skeleton (Maldonado et al., 2020). The formation of longer spicules (i.e. 250-300  $\mu$ m), known as megascleres, is completed outside the cell, where silica is deposited externally (Schröder et al., 2007)."

-Line 200: We corrected the sentence as "Diatoms are characterised by their ability to exploit silicic acid to build a siliceous external cell wall, called a frustule".

- Line 438: Removed the following section as the numbers are introduced again in section 4: "Their ecological success marks diatoms as important contributors of biogeochemical cycles. Globally, diatoms produce approximately 255 Tmol Si  $yr^{-1}$  of bSi, with regional variations influenced by nutrient availability and oceanic regimes (Tréguer et al., 2021). Coastal areas exhibit silica production rates 3 - 12 times higher than the global average, while oligotrophic gyres show rates 2 - 4 times lower (Brzezinski et al., 1998). "

Use of stable silicon isotopes for investigating silicification

Whilst stable silicon isotopes as a tool for understanding ocean silicon cycling are mentioned throughout, there are also studies that have used silicon isotopes either specifically (or as part of a discussion) to investigate silicification processes. I would suggest it could be useful for the authors to include some examples here. For example, Marron et al., 2019, investigated silicification in cultured choanoflagellates using silicon isotopes, using isotopes to probe the possibility of similar silicification pathways between choanoflagellates and their close relatives, sponges (c.f. mechanistic modelling of sponge silicon isotope fractionation during silicification in Wille et al, 2010 – already cited, updated recently by Maldonado & Hendry, 2025). In addition to culture and field studies, Cassarino et al. 2021 investigated the role of diatom proteins in fractionating silicon isotopes using in vitro experiments using the R5 peptide, which is a model for the natural biomolecule known to play a role in diatom silicification.

**AUTHORS' RESPONSE:**

We have revised box 1 "Studying biosilicification and silicon uptake: from molecular tools to radioisotope enrichment physiological experiments" to include examples where silicon isotopes have been directly applied to investigate silicification pathways, including studies on choanoflagellates (Marron et al., 2019), sponges (Wille et al., 2010; Maldonado & Hendry, 2025), and in vitro experiments with the R5 peptide (Cassarino et al., 2021). These additions strengthen the link between isotopic techniques and the mechanistic understanding of biosilicification.

**Minor comments:**

Line 88 refers to 32Si studies, but there are very few references to their utilisation throughout (just mentioned briefly in Box 1). There could be some pertinent studies cited for investigating silicon utilisation of silicification, for example (e.g., Krause et al., 2019 – already cited), perhaps in Section 4.

**AUTHORS' RESPONSE:**

We appreciate the reviewer's comment. For reasons of readability, we decided to limit the number of references cited in-text to a maximum of three per example. This is why it seems only briefly cited in box 1. However, as suggested, we have added additional references in Section 4 to highlight studies that employed silicon radioisotopes 32Si to investigate silicon uptake and biogenic silica production. This strengthens the coverage of radioisotope applications while keeping the reference list concise in other parts of the manuscript.

A minor point but – whilst I understand that it could be useful to include plant silicification for completeness – I do wonder about the relevance for a paper that's specifically about marine processes. E.g., could the paragraph starting on line 249 be trimmed? And Section 3.6. could also be trimmed. Especially given the lack of information about silicification in seagrasses (e.g., line 277, line 550 onwards)?

**AUTHORS' RESPONSE:**

We thank the Reviewer for this comment. We collectively decided that including some information about plant silicification and plant ecology could be useful, in order to stimulate interactions between different disciplines (e.g., BOX 3). In particular, some concepts and methods could be easily transferred from one discipline to another, but it is still poorly done because of science fragmentation.

In addition to this conceptual reason, bSi production by seagrass  $(0.58-3.33\times10^{-3} \,\mathrm{Tmol}\text{-Si yr}^{-1})$  and macroalgae  $(0.27-1.33 \,\mathrm{Tmol}\text{-Si yr}^{-1})$  in the world ocean could actually be similar to the global bSi production of sponges and benthic diatoms (please see section 4.2.1). For this reason, neglecting plants in the marine Si cycling could lead to significant discrepancies.

That being said, we agree with the Reviewer that sections on plant silicification and ecology might be too long for a paper mostly about marine ecology and biogeochemistry. As such, we made the following changes.

-Only the following sentences were kept in the plant silicification section (L. 247-250):

"In plants, it is increasingly accepted that some cell wall polymers and proteins can initiate and control silicification (Zexer et al., 2023). For instance, the deposits of silica in the so-called silica cells found in many grass species are thought to be controlled by a protein named Siliplant1 (Kumar et al., 2020b; Zexer et al., 2023). These works have been pivotal to further understanding plant silicification and its potential links to other silicifying organisms (Kumar et al., 2020a)."

-We also trimmed section 3.6, and in particular, the parts on terrestrial aquatic and wetland plants. However, we kept the paragraph on seagrasses (L. 547 to L.558) as it is.

Section 3: The section on diversity in silicifiers in the ocean is very useful and very pertinent for those embarking on functional trait models that include silicic acid and silicifiers (e.g., Naidoo-Bagwell et al., 2024). This is alluded to on line 74, but perhaps the authors could mention improved model predictive power as a motivation for section 3 in its introduction (e.g., could make a reference to Box 3)?

**AUTHORS' RESPONSE:**

We have revised the introduction to Section 3 to explicitly highlight that documenting the diversity of silicifiers not only improves ecological understanding but also strengthens functional trait and biogeochemical models by enhancing their predictive power. To illustrate this point, we now refer to Naidoo-Bagwell et al. (2024), who demonstrated improved model performance by including a diatom functional group requiring dissolved silica in the Earth system model cGEnIE, and we link this discussion to Box 3.

Section 4.3.2. The authors do a great job in summarizing the discussion surrounding the addition of 'new' silicon into the ocean via the in situ or benthic dissolution of lithogenic particulates (e.g., Fabre et al., 2019 – already cited). There is also evidence, which could be included in this section (alluded to on line 795), that lithogenic dissolution contributes significantly to porewater dSi accumulation and flux (e.g., Ward et al., 2022). What does this mean for our understanding of the role of bSI dissolution in benthic silicon cycling, and closing silicon budgets?

**AUTHORS' RESPONSE:**

Thank you for bringing our attention to this study. It is indeed very relevant. It has now been cited in the relevant section of the text. We have also added the following sentence: "Overall, the importance of the lithogenic and authigenic Si phase in modulating benthic Si fluxes is being increasingly recognised as the global marine Si cycle is described with increasing fidelity." (paragraph 2 of Section 4.3.2)

I also wonder if more could be emphasised here about the role of water column recycling and exchange processes. For example, there is evidence that detrital (biological or, indeed, lithogenic) material dissolution releases silicon that could support biological production (e.g., et al., 2019; Ng et al., 2024).

**AUTHORS' RESPONSE:**

Thank you for this suggestion. We have now added the Ng et al. reference to the relevant portion of the text discussing glaciated environments (section 4.3.3 paragraph 2). In order to maintain focus, we are unable to include more extensive review of water column processes and diatom productivity in this section, given its focus on the three different interfaces.

Section 4.3.3. Again a useful section, but could link the points summarised here more clearly to silicifiers to make sure it's relevant for the paper's key theme.

**AUTHORS' RESPONSE:**

In Section 4.3.3, we have revised the text to more explicitly link cryosphere-related silicon fluxes and processes to their relevance for silicifying organisms. For example, we now highlight how glacially derived amorphous silica can fuel diatom blooms in coastal systems, how biogenic silica recycling within sea ice sustains polar diatom communities, and how permafrost-driven shifts in silicon sources may affect the availability and isotopic composition of silica for Arctic silicifiers. These additions strengthen the connection of this section to the central theme of the paper.

**Additional references:**

Cassarino, L., et al. (2021). A biomimetic peptide has no effect on the isotopic fractionation during in vitro silica precipitation. Scientific reports, 11(1), 1-10.

Marron, A., et al. (2019). The silicon isotopic composition of choanoflagellates: implications for a mechanistic understanding of isotopic fractionation during biosilicification. Biogeosciences. 16(24), 4805-4813.

Maldonado, M., & Hendry, K. R. (2025). Revisiting the silicon isotopic signal of sponge skeletons and its implications. Limnology and Oceanography. https://doi.org/10.1002/lno.70138

Naidoo-Bagwell, A. A., et al. (2024). A diatom extension to the cGEnIE Earth system model—EcoGEnIE 1.1. Geoscientific Model Development, 17(4), 1729-1748.

Ng, H. C., et al. (2024). Detrital input sustains diatom production off a glaciated Arctic coast. Geophysical Research Letters, 51(12), e2024GL108324.

Livage, J. (2018). Bioinspired nanostructured materials. Comptes Rendus. Chimie, 21(10), 969-973.

Ward, J. P., et al. (2022). Benthic silicon cycling in the Arctic Barents Sea: A reaction-transport model study. Biogeosciences Discussions, 2022, 1-3

---

## Author Comment (AC2)

**Response to Reviewer #2**

The authors aim to synthetize current understanding in biosilification across diverse organisms and its influence on the global marine Si cycle.

In this manuscript, without a doubt, the authors have achieved their goal, delivering to a large audience existing knowledge on the different processes that control silicification in various organisms, exploring the taxonomic diversity of marine silicifiers, and assesing their role in Si cycling across geological timescales and in the modern ocean.

The four figures are helpful to illustrate silicification processes in unicellular and multicellular organisms, the size range of silicifiers, the processs that control the Si cycle in the modern global ocean, and the schematic of the silicon cycle and isotopic fractionation during various processes, using diatoms as a model for the biogenic silica component (included in box 4).

Four boxes detail the use of molecular tools and radioactive tracers for studying biosilificication processes and silicon uptake, that of siliceous microfossils as markers of past and present oceanic environments, trait-based approaches to understand the function of silicification in diatoms and plants, and the use of silicon natural stable isotopes to trace the marine silica cycle. Interstingly, this manuscript also shed light on silicification processes in terrestrial organisms and compare them to those occurring in marine organisms.

**AUTHORS' RESPONSE:**

We would like to warmly thank Reviewer 2 for their thorough and encouraging evaluation of our manuscript "Silicification in the ocean: from molecular pathways to silicifiers' ecology and biogeochemical cycles". We greatly appreciate your recognition of our efforts to synthesize the current understanding of biosilicification across diverse organisms and its influence on the marine silicon cycle.

We are pleased that you found the manuscript comprehensive and valuable. Your positive assessment of the figures and boxes, and the way they illustrate processes and tools relevant to silicification, is particularly encouraging. We are also grateful that you highlighted our inclusion of comparisons between marine and terrestrial silicification processes, as this was an aspect we aimed to emphasize in order to broaden the perspective for readers.

In the following, we provide detailed responses to your specific comments and suggestions for improvement, and we explain the changes made in the revised manuscript.

**General comment:**

Page 62 line 1991: It seems that Maria Lopez-Acosta and Antonia U. Thielecke are acting as the coordinators of the writing group. If this is correct their names should come first.

**AUTHORS' RESPONSE:**

We thank the reviewer for their thoughtfulness. However, responsibilities were distributed equally and corresponding author is not an indicator of a higher contribution.

-Line 2000: The writing and coordination of this manuscript was very much a group effort and responsibilities were distributed equally among the participants. Hence, neither the order of authors nor the role of corresponding authors is a reflection of their contribution to the work.

Minor comments:

Page 2 line 4: what is the meaning of « regulators »?

**AUTHORS' RESPONSE:**

We have now rephrased this sentence for clarity.

Page 4 line 103: regarding this topic Jacques Livage's works should be mentioned, for instance his publication on « Bioinspired nanostructured materials », published in September 2018 in Comptes Rendus Chimie 21(10) DOI:10.1016/j.crci.2018.08.001

**AUTHORS' RESPONSE:**

Thank you for suggesting this reference. It has been added to this section at the end of this sentence: "This stark difference highlights the efficiency of biological silicification, which occurs under ambient environmental conditions without the need for high temperatures or external catalysts (Livage, 2018)."

Page 6 line 157, EGXQ and GRQ: meaning?

**AUTHORS' RESPONSE:**

This sentence has been rearranged to clarify that these are amino acid motifs. The letters are commonly used abbreviations for amino acids (E: glutamic acid, G: glycine, Q: glutamine, R: arginine), while X usually denotes an unknown amino acid or nonconsensus between different protein sequences.

Page 13 line 391: how many species of diatoms?

**AUTHORS' RESPONSE:**

Thank you. We added the current number of described species and the total estimate to the text.

Page 16 line 465: Which recent studies?

**AUTHORS' RESPONSE:**

We added the appropriate reference to this study (Laget et al., 2024).

Page 16 line 492: could contribute up to 20% of the bSi production of the global ocean.

**AUTHORS' RESPONSE:**

Thank you. We adjusted the sentence accordingly.

Page 20 Figure 3 should be improved. Aeolian inputs and reverse weathering are missing processes (see Tréguer et al. 2021). At a first look the reader gets the feeling that sponges and glaciers/Ice sheets play a major role in the Si global marine cycle, which is not true.

**AUTHORS' RESPONSE:**

Thank you for the feedback. We adjusted the figure to include weathering and aeolian input as processes and reduced the size of the ice and the number of the sponges to more accurately reflect their contribution to the Si cycle. We also note that the figure does emphasize the specific processes and interfaces that we focus on in our manuscript, and is not meant to be a proportional depiction of their global importance. Regarding "reverse weathering", it is in fact included under the processes of "Alteration" and "Precipitation" in the figure, since it is a specific type of diagenetic alteration/precipitation, as discussed on L838-852.

Page 23 line 685. Figure 3 of Tréguer and De La Rocha 2013 or Figure 1 of this manuscript?

**AUTHORS' RESPONSE:**

Thank you for the comment. We revised the sentence as "Silicon, which is ultimately derived from the weathering of the Earth's crust, is transported to the ocean via various pathways (Fig. 3 and Table 1)".

Page 24 Table 1. Move Marine Si sinks on page 25

**AUTHORS' RESPONSE:**

-Table 1:

We moved "Marine Si sinks" on page 25 of Table 1, but after adjusting the content of Table 1, it is then moved to page 24. We thank you for the comment.

Page 25 Table 1 (continuation): Marine geological residence time

**AUTHORS' RESPONSE:**

-Table 1:

We changed "Marine Si residence Time (yrs)" as "Marine geological residence time of Si (yrs)"

Page 25 line 715. ...this task is challenging. Impossible is too much.

**AUTHORS' RESPONSE:**

-Line 715:

We deleted the phrase "if not impossible,"

Page 26 line 730 submarine groundwater (Cho et al. 2018; Rahman et al., 2019;...)

**AUTHORS' RESPONSE:**

-Line 728:

Thank you for this suggestion. We added the "Cho et al., 2018" which is a very supportive reference.

Page 27 line 750 and followings. The authors should say a few words about the impacts of the Yangtze

huge dam on the dowstream ecosystems as regards the Si cycle.

**AUTHORS' RESPONSE:**

Thank you for the suggestion, however to maintain the focus of the manuscript, we only discuss global riverine inputs only broadly as a whole, and we feel discussing specific rivers or indeed the continental

Si cycle in more detail is unfortunately outside of the scope of this manuscript.

Page 28 line 797 and following. The authors should mention Luo et al. (2022)'s work on in the deepest hadal trench (Marianna) sediments, where alkalinity profiles are generated by organic matter

mineralization that represents the dominant early diagenetic process in marine sediments, despite the

occurrence of reserve weathering.

**AUTHORS' RESPONSE:**

Thank you for the suggestion, we have now included references to Luo et al. 2022, as well as 2025 in the sentences discussing authigenic precipitation and the coupling to organic matter, respectively.

(Paragraph 2 of section 4.3.2)

Page 29 line 825: aSi definition.

**AUTHORS' RESPONSE:**

-Line 825:

Thank you for the comment. The definition of aSi was given in Line 819: amorphous silica (aSi).

Page 29 line 834: « CNSi » as an abreviation of colloidal-nanoparticulate size Si is not pertinent (CNSi

= carbon, nitrogen, silicon.

**AUTHORS' RESPONSE:**

-Line 832:

We deleted the abbreviation "CNSi" as it was only used once in the manuscript.

Page 35 line 1031: lower case letters.

**AUTHORS' RESPONSE:**

-Line 1028-1029:

We changed the upper-case letters into lower-case.

4

Page 64 Box 2 last line: « only about 10% of surface-dwelling diatoms are preserved in sediments ». This seems too high. What are the references supporting this statement?

**AUTHORS' RESPONSE:**

We modified this to "in about 1-10% reach the seafloor and are preserved in the sediment (Crosta and Koc, 2007)".

Page 67 Box 4 line 12 « isotopic fractionation in sponges' spicules correlates closely with ambient dSi concentrations »... is this statement in agreement with Maldonado and Hendry (L&O, July 2025)?

**AUTHORS' RESPONSE:**

Thank you for this comment. A sentence has been added after this statement that incorporates the results from Maldonado and Hendry, 2025, into this section